# FEDPARA: LOW-RANK HADAMARD PRODUCT FOR COMMUNICATION-EFFICIENT FEDERATED LEARNING

**Nam Hyeon-Woo**[1]**, Moon Ye-Bin**[1]**, Tae-Hyun Oh**[1,2,3]
[1]Department of Electrical Engineering, POSTECH    [2]Graduate School of AI, POSTECH
[3]Yonsei University
{hyeonw.nam, ybmoon, taehyun}@postech.ac.kr

## ABSTRACT

In this work, we propose a communication-efficient parameterization, FedPara, for federated learning (FL) to overcome the burdens on frequent model uploads and downloads. Our method re-parameterizes weight parameters of layers using low-rank weights followed by the Hadamard product. Compared to the conventional low-rank parameterization, our FedPara method is not restricted to low-rank constraints, and thereby it has a far larger capacity. This property enables to achieve comparable performance while requiring 3 to 10 times lower communication costs than the model with the original layers, which is not achievable by the traditional low-rank methods. The efficiency of our method can be further improved by combining with other efficient FL optimizers. In addition, we extend our method to a personalized FL application, pFedPara, which separates parameters into global and local ones. We show that pFedPara outperforms competing personalized FL methods with more than three times fewer parameters. Project page: https://github.com/South-hw/FedPara_ICLR22

## 1 INTRODUCTION

Federated learning (FL; McMahan et al., 2017) has been proposed as an efficient collaborative learning strategy along with the advance and spread of mobile and IoT devices. FL allows leveraging local computing resources of edge devices and locally stored private data without data sharing for privacy. FL typically consists of the following steps: (1) clients download a globally shared model from a central server, (2) the clients locally update each model using their own private data without accessing the others' data, (3) the clients upload their local models back to the server, and (4) the server consolidates the updated models and repeats these steps until the global model converges. FL has the key properties (McMahan et al., 2017) that differentiate it from distributed learning:

- **Heterogeneous data.** Data is decentralized and non-IID as well as unbalanced in its amount due to different characteristics of clients; thus, local data does not represent the population distribution.

- **Heterogeneous systems.** Clients consist of heterogeneous setups of hardware and infrastructure; hence, those connections are not guaranteed to be online, fast, or cheap. Besides, massive client participation is expected through different communication paths, causing communication burdens.

These FL properties introduce challenges in the convergence stability with heterogeneous data and communication overheads. To improve the stability and reduce the communication rounds, the prior works in FL have proposed modified loss functions or model aggregation methods (Li et al., 2020; Karimireddy et al., 2020; Acar et al., 2021; Yu et al., 2020; Reddi et al., 2021). However, the transferred data is still a lot for the edge devices with bandwidth constraints or countries having low-quality communication infrastructure.[1] A large amount of transferred data introduces an energy consumption issue on edge devices because wireless communication is significantly more power-intensive than computation (Yadav & Yadav, 2016; Yan et al., 2019).

---

[1]The gap between the fastest and the lowest communication speed across countries is significant; approximately 63 times different (Speedtest).

In this work, we propose a communication-efficient re-parameterization for FL, `FedPara`, which reduces the number of bits transferred per round. `FedPara` directly re-parameterizes each fully-connected (FC) and convolutional layers of the model to have a small and factorized form while preserving the model's capacity. Our key idea is to combine the Hadamard product with low-rank parameterization as $\mathbf{W} = (\mathbf{X}_1 \mathbf{Y}_1^\top) \odot (\mathbf{X}_2 \mathbf{Y}_2^\top) \in \mathbb{R}^{m \times n}$, called *low-rank Hadamard product*. When $\text{rank}(\mathbf{X}_1 \mathbf{Y}_1^\top) = \text{rank}(\mathbf{X}_2 \mathbf{Y}_2^\top) = r$, then $\text{rank}(\mathbf{W}) \leq r^2$. This outstanding property facilitates spanning a full-rank matrix with much fewer parameters than the typical $m \times n$ matrix. It significantly reduces the communication burdens during training. At the inference phase, we pre-compose and maintain $\mathbf{W}$ that boils down to its original structure; thus, `FedPara` does not alter computational complexity at inference time. Compared to the aforementioned prior works that tackle reducing the required communication rounds for convergence, our `FedPara` is an orthogonal approach in that `FedPara` does not change the optimization part but re-defines each layer's internal structure.

We demonstrate the effectiveness of `FedPara` with various network architectures, including VGG, ResNet, and LSTM, on standard classification benchmark datasets for both IID and non-IID settings. The accuracy of our parameterization outperforms that of the traditional low-rank parameterization baseline given the same number of parameters. Besides, `FedPara` has comparable accuracy to original counterpart models and even outperforms as the number of parameters increases at times. We also combine `FedPara` with other FL algorithms to improve communication efficiency further.

We extend `FedPara` to the personalized FL application, named `pFedPara`, which separates the roles of each sub-matrix into global and local inner matrices. The global and local inner matrices learn the globally shared common knowledge and client-specific knowledge, respectively. We devise three scenarios according to the amount and heterogeneity of local data using the subset of FEMNIST and MNIST. We demonstrate performance improvement and robustness of `pFedPara` against competing algorithms. We summarize our main contributions as follows:

- We propose `FedPara`, a low-rank Hadamard product parameterization for communication-efficient FL. Unlike traditional low-rank parameterization, we show that `FedPara` can span a full-rank matrix and tensor with reduced parameters. We also show that `FedPara` requires up to ten times fewer total communication costs than the original model to achieve target accuracy. `FedPara` even outperforms the original model by adjusting ranks at times.

- Our `FedPara` takes a novel approach; thereby, our `FedPara` can be combined with other FL methods to get mutual benefits, which further increase accuracy and communication efficiency.

- We propose `pFedPara`, a personalization application of `FedPara`, which splits the layer weights into global and local parameters. `pFedPara` shows more robust results in challenging regimes than competing methods.

## 2 METHOD

In this section, we first provide the overview of three popular low-rank parameterizations in Section 2.1 and present our parameterization, `FedPara`, with its algorithmic properties in Section 2.2. Then, we extend `FedPara` to the personalized FL application, `pFedPara`, in Section 2.3.

**Notations.** We denote the Hadamard product as $\odot$, the Kronecker product as $\otimes$, $n$-mode tensor product as $\times_n$, and the $i$-th unfolding of the tensor $\mathcal{T}^{(i)} \in \mathbb{R}^{k_i \times \prod_{j \neq i} k_j}$ given a tensor $\mathcal{T} \in \mathbb{R}^{k_1 \times \cdots \times k_n}$.

### 2.1 OVERVIEW OF LOW-RANK PARAMETERIZATION

The low-rank decomposition in neural networks has been typically applied to pre-trained models for compression (Phan et al., 2020), whereby the number of parameters is reduced while minimizing the loss of encoded information. Given a learned parameter matrix $\mathbf{W} \in \mathbb{R}^{m \times n}$, it is formulated as finding the best rank-$r$ approximation, as $\arg\min_{\widetilde{\mathbf{W}}} \|\mathbf{W} - \widetilde{\mathbf{W}}\|_F$ such that $\widetilde{\mathbf{W}} = \mathbf{X}\mathbf{Y}^\top$, where $\mathbf{X} \in \mathbb{R}^{m \times r}$, $\mathbf{Y} \in \mathbb{R}^{n \times r}$, and $r \ll \min(m, n)$. It reduces the number of parameters from $O(mn)$ to $O((m + n)r)$, and its closed-form optimal solution can be found by SVD.

This matrix decomposition is applicable to the FC layers and the reshaped kernels of the convolution layers. However, the natural shape of a convolution kernel is a fourth-order tensor; thus, the low-rank tensor decomposition, such as Tucker and CP decomposition (Lebedev et al., 2015; Phan et al.,

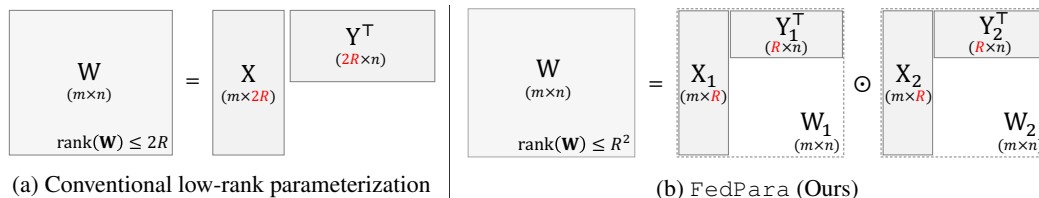

(a) Conventional low-rank parameterization      (b) `FedPara` (Ours)

Figure 1: Illustrations of low-rank matrix parameterization and `FedPara` with the same number of parameters $2R(m+n)$. (a) Low-rank parameterization is the summation of $2R$ number of rank-1 matrices, $\mathbf{W} = \mathbf{X}\mathbf{Y}^\top$, and $\mathrm{rank}(\mathbf{W}) \leq 2R$. (b) `FedPara` is the Hadamard product of two low-rank inner matrices, $\mathbf{W} = \mathbf{W}_1 \odot \mathbf{W}_2 = (\mathbf{X}_1\mathbf{Y}_1^\top) \odot (\mathbf{X}_2\mathbf{Y}_2^\top)$, and $\mathrm{rank}(\mathbf{W}) \leq R^2$.

| Layer | Parameterization | # Params. | Maximal Rank | Example [# Params. / Rank] |
|---|---|---|---|---|
| FC Layer | Original | $mn$ | $\min(m, n)$ | 66 K / 256 |
| | Low-rank | $2R(m+n)$ | $2R$ | 16 K / 32 |
| | `FedPara` | $2R(m+n)$ | $R^2$ | 16 K / 256 |
| Convolutional Layer | Original | $OIK_1K_2$ | $\min(O, IK_1K_2)$ | 590 K / 256 |
| | Low-rank | $2R(O + I + RK_1K_2)$ | $2R$ | 21 K / 32 |
| | `FedPara` (Proposition 1) | $2R(O + IK_1K_2)$ | $R^2$ | 82 K / 256 |
| | `FedPara` (Proposition 3) | $2R(O + I + RK_1K_2)$ | $R^2$ | 21 K / 256 |

Table 1: The number of parameters, maximal rank, and example. We assume that the weights of the FC and convolutional layers are in $\mathbb{R}^{m \times n}$ and $\mathbb{R}^{O \times I \times K_1 \times K_2}$, respectively. The rank of the convolutional layer is the rank of the $1^{\text{st}}$ unfolding tensor. As a reference example, we set $m = n = O = I = 256$, $K_1 = K_2 = 3$, and $R = 16$.

2020), can be a more suitable approach. Given a learned high-order tensor $\mathcal{T} \in \mathbb{R}^{k_1 \times \cdots \times k_n}$, Tucker decomposition multiplies a kernel tensor $\mathcal{K} \in \mathbb{R}^{r_1 \times \cdots \times r_n}$ with matrices $\mathbf{X}_i \in \mathbb{R}^{r_i \times n_1}$, where $r_i = \mathrm{rank}(\widetilde{\mathcal{T}}^{(i)})$ as $\widetilde{\mathcal{T}} = \mathcal{K} \times_1 \mathbf{X}_1 \times_2 \cdots \times_n \mathbf{X}_n$, and CP decomposition is the summation of rank-1 tensors as $\widetilde{\mathcal{T}} = \sum_{i=1}^{i=r} \mathbf{x}_i^1 \times \mathbf{x}_i^2 \times \cdots \times \mathbf{x}_i^n$, where $\mathbf{x}_i^j \in \mathbb{R}^{k_j}$. Likewise, it also reduces the number of model parameters. We refer to these rank-constrained structure methods simply as conventional low-rank constraints or low-rank parameterization methods.

In the FL context, where the parameters are frequently transferred between clients and the server during training, the reduced parameters lead to communication cost reduction, which is the main focus of this work. The post-decomposition approaches (Lebedev et al., 2015; Phan et al., 2020) using SVD, Tucker, and CP decompositions do not reduce the communication costs because those are applied to the original parameterization after finishing training. That is, the original large-size parameters are transferred during training in FL, and the number of parameters is reduced after finishing training.

We take a different notion from the low-rank parameterizations. In the FL scenario, we train a model from scratch with low-rank constraints, but specifically with *low-rank Hadamard product re-parameterization*. We re-parameterize each learnable layer, including FC and convolutional layers, and train the surrogate model by FL. Different from the existing low-rank method in FL (Konečný et al., 2016), our parameterization can achieve comparable accuracy to the original counterpart.

## 2.2 FedPara: A Communication-Efficient Parameterization

As mentioned, the conventional low-rank parameterization has limited expressiveness due to its low-rank constraint. To overcome this while maintaining fewer parameters, we present our new low-rank Hadamard product parameterization, called `FedPara`, which has the favorable property as follows:

**Proposition 1** *Let $\mathbf{X}_1 \in \mathbb{R}^{m \times r_1}, \mathbf{X}_2 \in \mathbb{R}^{m \times r_2}, \mathbf{Y}_1 \in \mathbb{R}^{n \times r_1}, \mathbf{Y}_2 \in \mathbb{R}^{n \times r_2}, r_1, r_2 \leq \min(m, n)$ and the constructed matrix be $\mathbf{W} := (\mathbf{X}_1\mathbf{Y}_1^\top) \odot (\mathbf{X}_2\mathbf{Y}_2^\top)$. Then, $\mathrm{rank}(\mathbf{W}) \leq r_1 r_2$.*

All proofs can be found in the supplementary material including Proposition 1. Proposition 1 implies that, unlike the low-rank parameterization, a higher-rank matrix can be constructed using the Hadamard product of two inner low-rank matrices, $W_1$ and $W_2$ (Refer to Figure 1). If we choose the

inner ranks $r_1$ and $r_2$ such that $r_1 r_2 \geq \min(m, n)$, the constructed matrix does not have a low-rank restriction and is able to span a full-rank matrix with a high chance (See Figure 6); *i.e.*, `FedPara` has the minimal parameter property achievable to full-rank. In addition, we can control the number of parameters by changing the inner ranks $r_1$ and $r_2$, respectively, but we have the following useful property to set the hyper-parameters to be a minimal number of parameters with a maximal rank.

**Proposition 2** *Given $R \in \mathbb{N}$, $r_1 = r_2 = R$ is the unique optimal choice of the following criteria,*

$$\arg\min\nolimits_{r_1, r_2 \in \mathbb{N}} \quad (r_1 + r_2)(m + n) \quad \text{s.t.} \quad r_1 r_2 \geq R^2, \tag{1}$$

*and its optimal value is $2R(m + n)$.*

Equation 3 implies the criteria that minimize the number of weight parameters used in our parameterization with the target rank constraint of the constructed matrix as $R^2$. Proposition 2 provides an efficient way to set the hyper-parameters. It implies that, if we set $r_1 = r_2 = R$ and $R^2 \geq \min(m, n)$, `FedPara` is highly likely to have no low-rank restriction[2] even with much fewer parameters than that of a naïve weight, *i.e.*, $2R(m + n) \ll mn$. Moreover, given the same number of parameters, $\text{rank}(\mathbf{W})$ of `FedPara` is higher than that of the naïve low-rank parameterization by a square factor, as shown in Figure 1 and Table 1.

To extend Proposition 1 to the convolutional layers, we can simply reshape the fourth-order tensor kernel to the matrix as $\mathbb{R}^{O \times I \times K_1 \times K_2} \to \mathbb{R}^{O \times (IK_1K_2)}$ as a naïve way, where $O, I, K_1$, and $K_2$ are the output channels, the input channels, and the kernel sizes, respectively. That is, our parameterization spans convolution filters with a few basis filters of size $I \times K_1 \times K_2$. However, we can derive more efficient parameterization of convolutional layers without reshaping as follows:

**Proposition 3** *Let $\mathcal{T}_1, \mathcal{T}_2 \in \mathbb{R}^{R \times R \times k_3 \times k_4}$, $\mathbf{X}_1, \mathbf{X}_2 \in \mathbb{R}^{k_1 \times R}$, $\mathbf{Y}_1, \mathbf{Y}_2 \in \mathbb{R}^{k_2 \times R}$, $R \leq \min(k_1, k_2)$ and the convolution kernel be $\mathcal{W} := (\mathcal{T}_1 \times_1 \mathbf{X}_1 \times_2 \mathbf{Y}_1) \odot (\mathcal{T}_2 \times_1 \mathbf{X}_2 \times_2 \mathbf{Y}_2)$. Then, the rank of the kernel satisfies $\text{rank}(\mathcal{W}^{(1)}) = \text{rank}(\mathcal{W}^{(2)}) \leq R^2$.*

Proposition 3 is the extension of Proposition 1 but can be applied to the convolutional layer without reshaping. In the convolutional layer case of Table 1, given the specific tensor size, Proposition 3 requires 3.8 times fewer parameters than Proposition 1. Hence, we use Proposition 3 for the convolutional layer since the tensor method is more effective for common convolutional models.

Optionally, we employ non-linearity and the Jacobian correction regularization, of which details can be found in the supplementary material. These techniques improve the accuracy and convergence stability but not essential. Depending on the resources of devices, these techniques can be omitted.

## 2.3 pFedPara: Personalized FL Application

In practice, data are heterogeneous and personal due to different characteristics of each client, such as usage times and habits. `FedPer` (Arivazhagan et al., 2019) has been proposed to tackle this scenario by distinguishing global and local layers in the model. Clients only transfer global layers (the top layer) and keep local ones (the bottom layers) on each device. The global layers learn jointly to extract general features, while the local layers are biased towards each user.

With our `FedPara`, we propose a personalization application, `pFedPara`, in which the Hadamard product is used as a bridge between the global inner weight $\mathbf{W}_1$ and the local inner weight $\mathbf{W}_2$. Each layer of the personalized model is constructed by $\mathbf{W} = \mathbf{W}_1 \odot (\mathbf{W}_2 + \mathbf{1})$, where $\mathbf{W}_1$ is transferred

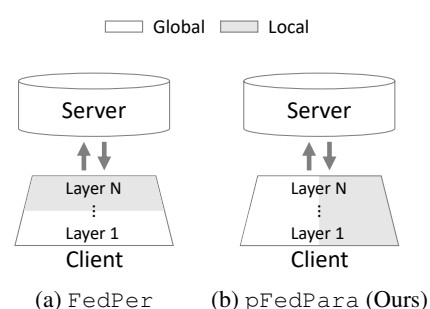

(a) `FedPer`    (b) `pFedPara` (Ours)

Figure 2: Diagrams of (a) `FedPer` and (b) `pFedPara`. The global part is transferred to the server and shared across clients, while local part remains private in each client.

[2]Its corollary and empirical evidence can be found in the supplementary material. Under Proposition 2, $R^2 \geq \min(m, n)$ is a necessary and sufficient condition for achieving a maximal rank.

to the server while $\mathbf{W}_2$ is kept in a local device during training. This induces $\mathbf{W}_1$ to learn globally shared knowledge implicitly and acts as a switch of the term $(\mathbf{W}_2 + \mathbf{1})$. Conceptually, we can interpret by rewriting $\mathbf{W} = \mathbf{W}_1 \odot \mathbf{W}_2 + \mathbf{W}_1 = \mathbf{W}_{per.} + \mathbf{W}_{glo.}$, where $\mathbf{W}_{per.} = \mathbf{W}_1 \odot \mathbf{W}_2$ and $\mathbf{W}_{glo.} = \mathbf{W}_1$. The construction of the final personalized parameter $\mathbf{W}$ in `pFedPara` can be viewed as an additive model of the global weight $\mathbf{W}_{glo.}$ and the personalizing residue $\mathbf{W}_{per.}$. `pFedPara` transfers only a half of the parameters compared to `FedPara` under the same rank condition; hence, the communication efficiency is increased further.

Intuitively, `FedPer` and `pFedPara` are distinguished by their respective split directions, as illustrated in Figure 2. We summarize our algorithms in the supplementary material. Although we only illustrate feed-forward network cases for convenience, it can be extended to general cases.

## 3 Experiments

We evaluate our `FedPara` in terms of communication costs, the number of parameters, and compatibility with other FL methods. We also evaluate `pFedPara` in three different non-IID scenarios. We use the standard FL algorithm, `FedAvg`, as a backbone optimizer in all experiments except for the compatibility experiments. More details and additional experiments can be found in the supplementary material.

### 3.1 Setup

**Datasets.** In `FedPara` experiments, we use four popular FL datasets: CIFAR-10, CIFAR-100 (Krizhevsky et al., 2009), CINIC-10 (Darlow et al., 2018), and the subset of Shakespeare (Shakespeare, 1994). We split the datasets randomly into 100 partitions for the CIFAR-10 and CINIC-10 IID settings and 50 partitions for the CIFAR-100 IID setting. For the non-IID settings, we use the Dirichlet distribution for random partitioning and set the Dirichlet parameter $\alpha$ as $0.5$ as suggested by He et al. (2020b). We assign one partition to each client and sample $16\%$ of clients at each round during FL. In `pFedPara` experiments, we use the subset of handwritten datasets: MNIST (LeCun et al., 1998) and FEMNIST (Caldas et al., 2018). For the non-IID setting with MNIST, we follow McMahan et al. (2017), where each of 100 clients has at most two classes.

**Models.** We experiment VGG and ResNet for the CNN architectures and LSTM for the RNN architecture as well as two FC layers for the multilayer perceptron. When using VGG, we use the `VGG16` architecture (Simonyan & Zisserman, 2015) and replace the batch normalization with the group normalization as suggested by Hsieh et al. (2020). `VGG16`$_{\text{ori.}}$ stands for the original `VGG16`, `VGG16`$_{\text{low}}$ the one with the low-rank tensor parameterization in a Tucker form by following TKD (Phan et al., 2020), and `VGG16`$_{\text{FedPara}}$ the one with our `FedPara`. In the `pFedPara` tests, we use two FC layers as suggested by McMahan et al. (2017).

**Rank Hyper-parameter.** We adjust the inner rank of $\mathbf{W}_1$ and $\mathbf{W}_2$ as $r = (1-\gamma)r_{min} + \gamma r_{max}$, where $r_{min}$ is the minimum rank allowing `FedPara` to achieve a full-rank by Proposition 2, $r_{max}$ is the maximum rank such that the number of `FedPara`'s parameters do not exceed the number of original parameters, and $\gamma \in [0, 1]$. We fix the same $\gamma$ for all layers for simplicity.[3] Note that $\gamma$ determines the number of parameters.

### 3.2 Quantitative Results

**Capacity.** In Table 2, we validate the propositions stating that our `FedPara` achieves a higher rank than the low-rank parameterization given the same number of parameters. We train `VGG16`$_{\text{low}}$ and `VGG16`$_{\text{FedPara}}$ for the same target rounds $T$, and use $10.25\%$ and $10.15\%$ of the `VGG16`$_{\text{ori.}}$ parameters, respectively, to be comparable. As shown in Table 2a, `VGG16`$_{\text{FedPara}}$ surpasses `VGG16`$_{\text{low}}$ on all the IID and non-IID settings benchmarks with noticeable margins. The same tendency is observed in the recurrent neural networks as shown in Table 2b. We train `LSTM` on the subset of Shakespeare. Table 2b shows that `LSTM`$_{\text{FedPara}}$ has higher accuracy than `LSTM`$_{\text{low}}$, where the number of parameters is set to $19\%$ and $16\%$ of `LSTM`, respectively. This experiment evidences that our `FedPara` has a better model expressiveness and accuracy than the low-rank parameterization.

---

[3]The parameter $\gamma$ can be independently tuned for each layer in the model. Moreover, one may apply neural architecture search or hyper-parameter search algorithms for further improvement. We leave it for future work.

| (a) CNN Models | CIFAR-10 ($T = 200$) | | CIFAR-100 ($T = 400$) | | CINIC-10 ($T = 300$) | |
|---|---|---|---|---|---|---|
| | IID | non-IID | IID | non-IID | IID | non-IID |
| VGG16$_{low}$ | 77.62 | 67.75 | 34.16 | 30.30 | 63.98 | 60.80 |
| VGG16$_{FedPara}$ (Ours) | **82.88** | **71.35** | **45.78** | **43.94** | **70.35** | **64.95** |

| (b) RNN Model | Shakespeare ($T = 500$) | |
|---|---|---|
| | IID | non-IID |
| LSTM$_{low}$ | 54.59 | 51.24 |
| LSTM$_{FedPara}$ (Ours) | **63.65** | **51.56** |

Table 2: Accuracy comparison between low-rank parameterization and FedPara. (a) The accuracy VGG16$_{low}$ and VGG16$_{FedPara}$. We set the target rounds $T = 200$ for CIFAR-10, 400 for CIFAR-100, and 300 for CINIC-10. (b) The accuracy of LSTM$_{low}$ and LSTM$_{FedPara}$. We set the target rounds $T = 500$ for the Shakespeare dataset.

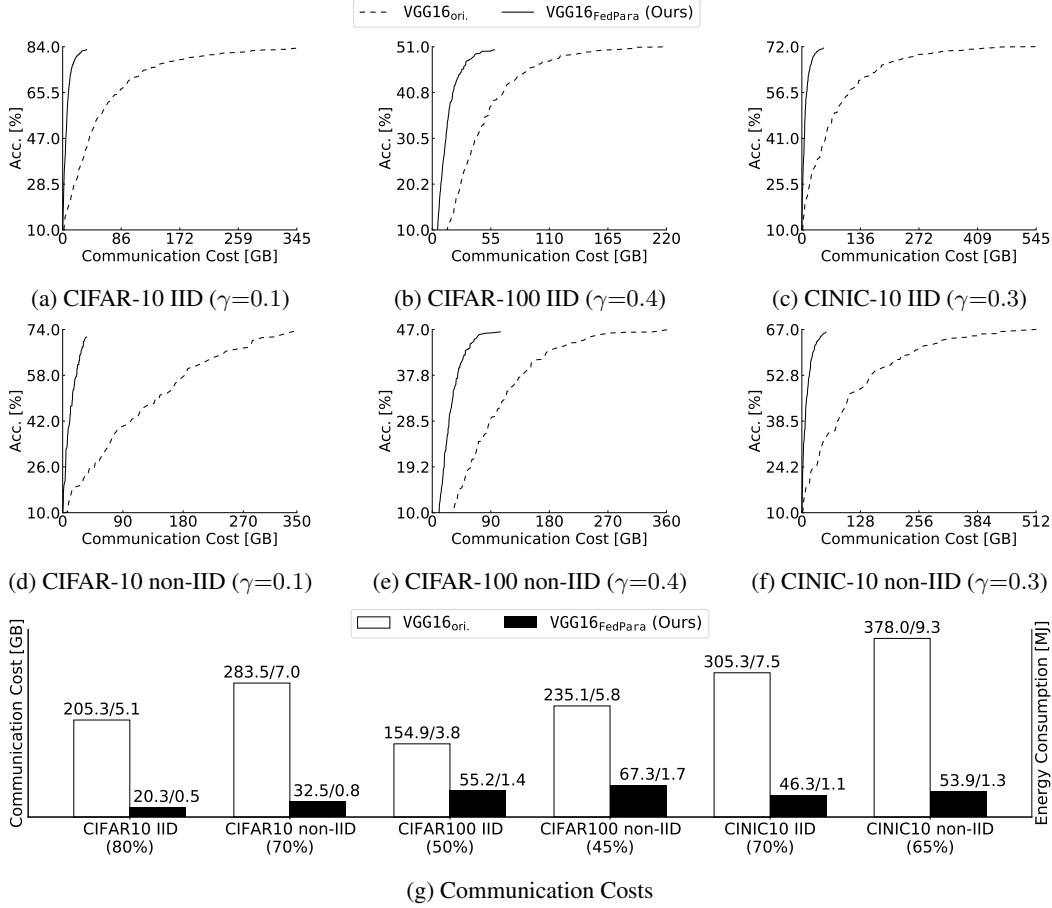

Figure 3: (a-f): Accuracy [%] ($y$-axis) vs. communication costs [GBytes] ($x$-axis) of VGG16$_{ori.}$ and VGG16$_{FedPara}$. Broken line and solid line represent VGG16$_{ori.}$ and VGG16$_{FedPara}$, respectively. (g): Size comparison of transferred parameters, which can be expressed as communication costs [GBytes] (left $y$-axis) or energy consumption [MJ] (right $y$-axis), for the same target accuracy. The white bars are the results of VGG16$_{ori.}$ and the black bars are the results of VGG16$_{FedPara}$. The target accuracy is denoted in the parentheses under the $x$-axis of (g).

**Communication Cost.** We compare VGG16$_{FedPara}$ and VGG16$_{ori.}$ in terms of accuracy and communication costs. FL evaluation typically measures the required rounds to achieve the target accuracy as communication costs, but we instead assess total transferred bit sizes, $2 \times$ (#participants)$\times$(model size)$\times$(#rounds), which considers up-/down-link and is a more practical communication cost metric. Depending on the difficulty of the datasets, we set the model size of VGG16$_{FedPara}$ as 10.1%, 29.4%, and 21.8% of VGG16$_{ori.}$ for CIFAR-10, CIFAR-100, and CINIC-10, respectively.

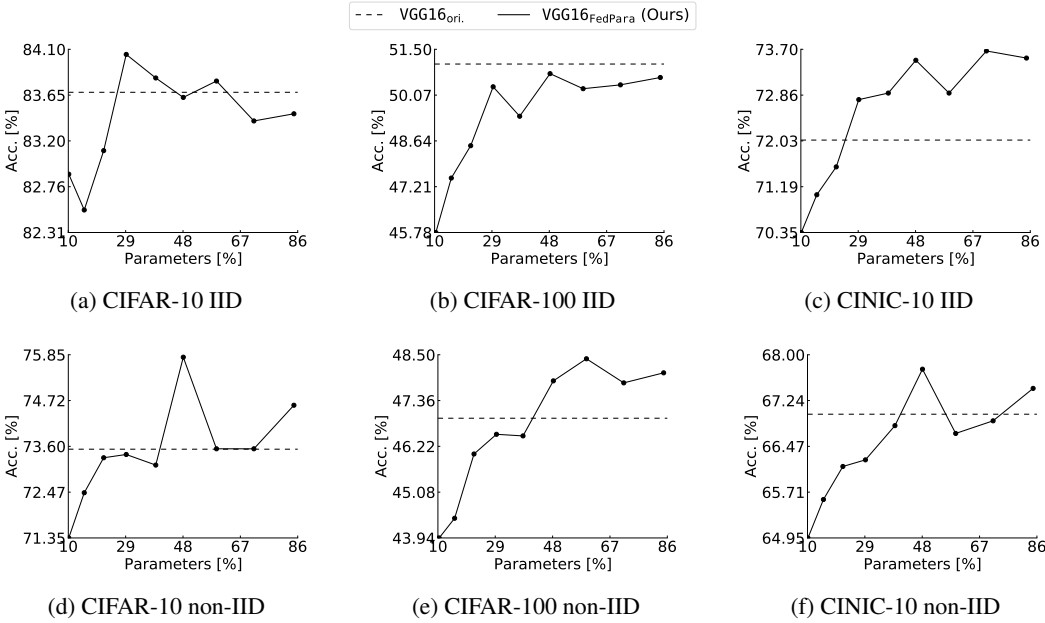

(a) CIFAR-10 IID      (b) CIFAR-100 IID      (c) CINIC-10 IID

(d) CIFAR-10 non-IID      (e) CIFAR-100 non-IID      (f) CINIC-10 non-IID

Figure 4: Test accuracy [%] ($y$-axis) vs. parameters ratio [%] ($x$-axis) of $VGG16_{FedPara}$ at the target rounds. The target rounds follow Table 2. The dotted line represents $VGG16_{ori.}$ with no parameter reduction, and the solid line $VGG16_{FedPara}$ adjusted by $\gamma \in [0.1, 0.9]$ in 0.1 increments.

|  | FedAvg | FedProx | SCAFFOLD | FedDyn | FedAdam |
|---|---|---|---|---|---|
| Accuracy ($T = 200$) | 82.88 | 78.95 | 84.72 | **86.05** | 82.48 |
| Round (80%) | 110 | - | 92 | **80** | 117 |

Table 3: The compatibility of `FedPara` with other FL algorithms. The first row is the accuracy of `FedPara` combined with other FL algorithms on the CIFAR-10 IID setting after 200 rounds, and the second row is the required rounds to achieve the target accuracy 80%.

In Figures 3a-3f, $VGG16_{FedPara}$ has comparable accuracy but requires much lower communication costs than $VGG16_{ori.}$. Figure 3g shows communication costs and energy consumption required for model training to achieve the target accuracy; we compute the energy consumption by the energy model of user-to-data center topology (Yan et al., 2019). $VGG16_{FedPara}$ needs 2.8 to 10.1 times fewer communication costs and energy consumption than $VGG16_{ori.}$ to achieve the same target accuracy. Because of these properties, `FedPara` is suitable for edge devices suffering from communication and battery consumption constraints.

**Model Parameter Ratio.** We analyze how the number of parameters controlled by the rank ratio $\gamma$ affects the accuracy of `FedPara`. As revealed in Figure 4, $VGG16_{FedPara}$'s accuracy mostly increases as the number of parameters increases. $VGG16_{FedPara}$ can achieve even higher accuracy than $VGG16_{ori.}$. It is consistent with the reports from the prior works (Luo et al., 2017; Kim et al., 2019) on model compression, where reduced parameters often lead to accuracy improvement, *i.e.*, regularization effects.

**Compatibility.** We integrate the `FedPara`-based model with other FL optimizers to show that our `FedPara` is compatible with them. We measure the accuracy during the target rounds and the required rounds to achieve the target accuracy. Table 3 shows that $VGG16_{FedPara}$ combined with the current state-of-the-art method, `FedDyn`, is the best among other combinations. Thereby, we can further save the communication costs by combining `FedPara` with other efficient FL approaches.

**Personalization.** We evaluate `pFedPara`, assuming no sub-sampling of clients for an update. We train two FC layers on the FEMNIST or MNIST datasets using four algorithms, `FedPAQ`, `FedAvg`, `FedPer`, and `pFedPara`, with ten clients. `FedPAQ` denotes the local models trained only using their own local data; `FedAvg` the global model trained by the `FedAvg` optimization; and `FedPer` and `pFedPara` the personalized models of which the first layer (`FedPer`) and half of the inner

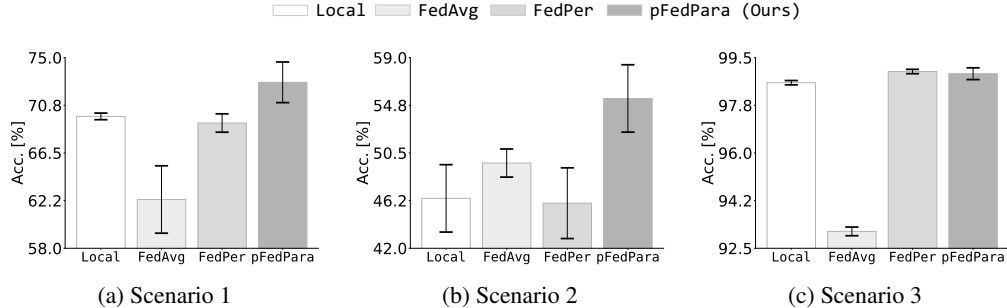

Figure 5: Average test accuracy over ten local models trained by each algorithm. (a) 100% of local training data on FEMNIST are used with the non-IID setting, which mimics enough local data to train and evaluates each local model on their own data. (b) 20% of local training data on FEMNIST are used with the non-IID setting, which mimics insufficient local data to train local models. (c) 100% of local training data on MNIST are used with the highly-skew non-IID setting, where each client has at most two classes. The error bars denote 95% confidence intervals obtained by 5 repeats.

matrices (`pFedPara`) are trained by sharing with the server, respectively, while the other parts are locally updated. We validate these algorithms on three scenarios in Figure 5.

In Scenario 1 (Figure 5a), the `FedPAQ` accuracy is higher than those of `FedAvg` and `FedPer` because each client has sufficient data. Nevertheless, `pFedPara` surpasses the other methods. In Scenario 2 (Figure 5b), the `FedAvg` accuracy is higher than that of `FedPAQ` because local data are too scarce to train the local models. The `FedPer` accuracy is also lower than that of `FedAvg` because the last layer of `FedPer` does not exploit other clients' data; thus, `FedPer` is susceptible to a lack of local data. The higher performance of `pFedPara` shows that our `pFedPara` can take advantage of the wealth of distributed data in the personalized setup. In Scenario 3 (Figure 5c), the `FedAvg` accuracy is much lower than the other methods due to the highly-skewed data distribution. In most scenarios, `pFedPara` performs better or favorably against the others. It validates that `pFedPara` can train the personalized models collaboratively and robustly.

Additionally, both `pFedPara` and `FedPer` save the communication costs because they partially transfer parameters; `pFedPara` transfers $3.4$ times fewer parameters, whereas `FedPer` transfers $1.07$ times fewer than the original model in each round. The reduction of `FedPer` is negligible because it is designed to transfer all the layers except the last one. Contrarily, the reduction of `pFedPara` is three times larger than `FedPer` because all the layers of `pFedPara` are factorized by the Hadamard product during training, and only a half of each layer's parameters are transmitted. Thus, `pFedPara` is far more suitable in terms of both personalization performance and communication efficiency in FL.

**Additional Experiments.** We conducted additional experiments of wall clock time simulation and other architectures including `ResNet`, `LSTM`, and `Pufferfish` (Wang et al., 2021), but the results can be found in the supplementary material due to the page limit.

## 4 RELATED WORK

**Federated Learning.** The most popular and de-facto algorithm, `FedAvg` (McMahan et al., 2017), reduces communication costs by updating the global model using a simple model averaging once a large number of local SGD iterations per round. Variants of `FedAvg` (Li et al., 2020; Yu et al., 2020; Diao et al., 2021) have been proposed to reduce the communication cost, to overcome data heterogeneity, or to increase the convergence stability. Advanced FL optimizers (Karimireddy et al., 2020; Acar et al., 2021; Reddi et al., 2021; Yuan & Ma, 2020) enhance the convergence behavior and improve communication efficiency by reducing the number of necessary rounds until convergence. Federated quantization methods (Reisizadeh et al., 2020; Haddadpour et al., 2021) combine the quantization algorithms with `FedAvg` and reduce only upload costs to preserve the model accuracy. Our `FedPara` is a drop-in replacement for layer parameterization, which means it is an orthogonal and compatible approach to the aforementioned methods; thus, our method can be integrated with other FL optimizers and model quantization.

**Distributed Learning.** In large-scale distributed learning of data centers, communication might be a bottleneck. Gradient compression approaches, including quantization (Alistarh et al., 2017; Bernstein et al., 2018; Wen et al., 2017; Reisizadeh et al., 2020; Haddadpour et al., 2021), sparsification (Alistarh et al., 2018; Lin et al., 2018), low-rank decomposition (Vogels et al., 2019; Wang et al., 2021), and adaptive compression (Agarwal et al., 2021) have been developed to handle communication traffic. These methods do not deal with FL properties such as data distribution and partial participants per round (Kairouz et al., 2019). Therefore, the extension of the distributed methods to FL is non-trivial, especially for optimization-based approaches.

**Low-rank Constraints.** As described in section 2.1, low-rank decomposition methods (Lebedev et al., 2015; Tai et al., 2016; Phan et al., 2020) are inappropriate for FL due to additional steps; the post-decomposition after training and fine-tuning. In FL, Konečnỳ et al. (2016) and Qiao et al. (2021) have proposed low-rank approaches. Konečnỳ et al. (2016) train the model from scratch with low-rank constraints, but the accuracy is degraded when they set a high compression rate. To avoid such degradation, FedDLR (Qiao et al., 2021) uses an ad hoc adaptive rank selection and shows the improved performance. However, once deployed, those models are inherently restricted by limited low-ranks. In particular, FedDLR requires matrix decomposition in every up/down transfer. In contrast, we show that FedPara has no such low-rank constraints in theory. Empirically, the models re-parameterized by our method show comparable accuracy to the original counterparts when trained from scratch.

# 5 DISCUSSION AND CONCLUSION

To overcome the communication bottleneck in FL, we propose a new parameterization method, FedPara, and its personalized version, pFedPara. We demonstrate that both FedPara and pFedPara can significantly reduce communication overheads with minimal performance degradation or better performance over the original counterpart at times. Even using a strong low-rank constraint, FedPara has no low-rank limitation and can achieve a full-rank matrix and tensor by virtue of our proposed low-rank Hadamard product parameterization. These favorable properties enable communication-efficient FL, of which regimes have not been achieved by the previous low-rank parameterization and other FL approaches. We conclude our work with the following discussions.

**Discussions.** FedPara conducts multiplications many times during training, including the Hadamard product, to construct the weights of the layers. These multiplications may potentially be more susceptible to gradient exploding, vanishing, dead neurons, or numerical instability than the low-rank parameterization with an arbitrary initialization. In our experiments, we have not observed such issues when using He initialization (He et al., 2015) yet. Investigating initializations appropriate for our model might improve potential instability in our method.

Also, we have discussed the expressiveness of each layers in neural networks in the view of rank. As another view of layer characteristics, statistical analysis of weights and activation also offers ways to initialize weights (He et al., 2015; Sitzmann et al., 2020) and understanding of neural networks (De & Smith, 2020), which is barely explored in this work. It would be a promising future direction to analyze statistical properties of composited weights from our parameterization and activations and may pave the way for FedPara-specific initialization or optimization.

Through the extensive experiments, we show the superior performance improvement obtained by our method, and it appears to be with no extra cost. However, the actual payoff exists in the additional computational cost when re-composing the original structure of $\mathbf{W}$ from our parameterization during training; thus, our method is slower than the original parameterization and low-rank approaches. However, the computation time is not a dominant factor in practical FL scenarios as shown in Table 7, but rather the communication cost takes a majority of the total training time. It is evidenced by the fact that FedPara offers better Pareto efficiency than all compared methods because our method has higher accuracy than low-rank approaches and much less training time than the original one. In contrast, the computation time might be non-negligible compared to the communication time in distributed learning regimes. While our method can be applied to distributed learning, the benefits of our method may be diminished there. Improving both computation and communication efficiency of FedPara in large-scale distributed learning requires further research. It would be a promising future direction.

## ETHICS STATEMENT

We describe the ethical aspect in various fields, such as privacy, security, infrastructure level gap, and energy consumption.

**Privacy and Security.** Although FL is privacy-preserving distributed learning, personal information may be leaked due to the adversary who hijacks the model intentionally during FL. Like other FLs, this risk is also shared with `FedPara` due to communication. Without care, the private data may be revealed by the membership inference or reconstruction attacks (Rigaki & Garcia, 2020). The local parameters in our `pFedPara` could be used as a private key to acquire the complete personal model, which would reduce the chance for the full model to be hijacked. It would be interesting to investigate whether our `pFedPara` guarantees privacy preserving or the way to improve robustness against those risks.

**Infrastructure Level Gap.** Another concern introduced by FL is limited-service access to the people living in countries having inferior communication infrastructure, which may raise human rights concerns including discrimination, excluding, *etc*. It is due to a larger bandwidth requirement of FL to transmit larger models. Our work may broaden the countries capable of FL by reducing required bandwidths, whereby it may contribute to addressing the technology gap between regions.

**Energy Consumption.** The communication efficiency of our `FedPara` directly leads to noticeable energy-saving effects in the FL scenario. It can contribute to reducing the battery consumption of IoT devices and fossil fuels used to generate electricity. Moreover, compared to the optimization-based FL approaches that reduce necessary communication rounds, our method allows more clients to participate in each learning round under the fixed bandwidth, which would improve convergence speed and accuracy further.

## ACKNOWLEDGEMENT

This work was supported by Institute of Information & communications Technology Planning & Evaluation (IITP) grant funded by the Korea government(MSIT) (No.2021-0-02068, Artificial Intelligence Innovation Hub), the National Research Foundation of Korea (NRF) grant funded by the Korea government (MSIT) (No. NRF-2021R1C1C1006799), and the "HPC Support" Project supported by the 'Ministry of Science and ICT' and NIPA.

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

## SUPPLEMENTARY

In this supplementary material, we present additional details, results, and experiments that are not included in the main paper due to the space limit. The contents of this supplementary material are listed as follows:

## CONTENTS

## A  MAXIMAL RANK PROPERTY

This section presents the proofs of our propositions and an additional analysis of our method's algorithmic behavior in terms of the rank property.

### A.1  PROOFS

**Proposition 1** *Let $\mathbf{X}_1 \in \mathbb{R}^{m \times r_1}, \mathbf{X}_2 \in \mathbb{R}^{m \times r_2}, \mathbf{Y}_1 \in \mathbb{R}^{n \times r_1}, \mathbf{Y}_2 \in \mathbb{R}^{n \times r_2}, r_1, r_2 \leq \min(m, n)$ and the constructed matrix be $\mathbf{W} := (\mathbf{X}_1 \mathbf{Y}_1^\top) \odot (\mathbf{X}_2 \mathbf{Y}_2^\top)$. Then, $\mathrm{rank}(\mathbf{W}) \leq r_1 r_2$.*

*Proof.* $\mathbf{X}_1 \mathbf{Y}_1^\top$ and $\mathbf{X}_2 \mathbf{Y}_2^\top$ can be expressed as the summation of rank-1 matrices such that $\mathbf{X}_i \mathbf{Y}_i^\top = \sum_{j=1}^{j=r_i} \mathbf{x}_{ij} \mathbf{y}_{ij}^\top$, where $\mathbf{x}_{ij}$ and $\mathbf{y}_{ij}$ are the $j$-th column vectors of $\mathbf{X}_i$ and $\mathbf{Y}_i$, and $i \in \{1, 2\}$. Then,

$$\mathbf{W} = \mathbf{X}_1 \mathbf{Y}_1^\top \odot \mathbf{X}_2 \mathbf{Y}_2^\top = \sum_{j=1}^{j=r_1} \mathbf{x}_{1j} \mathbf{y}_{1j}^\top \odot \sum_{j=1}^{j=r_2} \mathbf{x}_{2j} \mathbf{y}_{2j}^\top = \sum_{k=1}^{k=r_1} \sum_{j=1}^{j=r_2} (\mathbf{x}_{1k} \mathbf{y}_{1k}^\top) \odot (\mathbf{x}_{2j} \mathbf{y}_{2j}^\top). \quad (2)$$

$\mathbf{W}$ is the summation of $r_1 r_2$ number of rank-1 matrices; thus, $\mathrm{rank}(\mathbf{W})$ is bounded above $r_1 r_2$. $\square$

**Proposition 2** *Given $R \in \mathbb{N}$, $r_1 = r_2 = R$ is the unique optimal choice of the following criteria,*

$$\arg\min_{r_1, r_2 \in \mathbb{N}} \quad (r_1 + r_2)(m + n) \quad \text{s.t.} \quad r_1 r_2 \geq R^2, \quad (3)$$

*and its optimal value is $2R(m + n)$.*

*Proof.* We use arithmetic-geometric mean inequality and the given constraint. We have

$$(r_1 + r_2)(m + n) \geq 2\sqrt{r_1 r_2}(m + n) \geq 2R(m + n). \quad (4)$$

The equality holds if and only if $r_1 = r_2 = R$ by the arithmetic–geometric mean inequality. $\qquad \square$

**Corollary 1** *Under Proposition 2, $R^2 \geq \min(m, n)$ is a necessary and sufficient condition for achieving the maximal rank of $\mathbf{W} = (\mathbf{X}_1 \mathbf{Y}_1^\top) \odot (\mathbf{X}_2 \mathbf{Y}_2^\top) \in \mathbb{R}^{m \times n}$, where $\mathbf{X}_1 \in \mathbb{R}^{m \times r_1}, \mathbf{X}_2 \in \mathbb{R}^{m \times r_2}, \mathbf{Y}_1 \in \mathbb{R}^{n \times r_1}, \mathbf{Y}_2 \in \mathbb{R}^{n \times r2}$, and $r_1, r_2 \leq \min(m, n)$.*

*Proof.* We first prove the sufficient condition. Given $r_1 = r_2 = R$ under Proposition 2 and $R^2 \geq \min(m, n)$, $\mathrm{rank}(\mathbf{W}) \leq \min(r_1 r_2, m, n) = \min(R^2, m, n) = \min(m, n)$. The matrix $\mathbf{W}$ has no low-rank restriction; thus the condition, $R^2 \geq \min(m, n)$, is the sufficient condition.

The necessary condition is proved by contraposition; if $R^2 < \min(m, n)$, the matrix $\mathbf{W}$ cannot achieve the maximal rank. Since $r_1 = r_2 = R$ under Proposition 2 and $R^2 < \min(m, n)$, then $\mathrm{rank}(\mathbf{W}) \leq \min(r_1 r_2, m, n) = \min(R^2, m, n) = R^2 < \min(m, n)$. That is, $\mathrm{rank}(\mathbf{W})$ is upper-bounded by $R^2$, which is lower than the maximal achievable rank of $\mathbf{W}$. Therefore, the condition, $R^2 \geq \min(m, n)$, is the necessary condition because the contrapositive is true. $\qquad \square$

Corollary 1 implies that, with $R^2 \geq \min(m, n)$, the constructed weight $\mathbf{W}$ does not have the low-rank limitation, and allows us to define the minimum inner rank as $r_{min} := \min(\lceil \sqrt{m} \rceil, \lceil \sqrt{n} \rceil)$. If we set $r_1 = r_2 = r_{min}$, $\mathrm{rank}(\mathbf{W})$ of our `FedPara` can achieve the maximal rank because $r_1 r_2 = r_{min}^2 \geq \min(m, n)$ while minimizing the number of parameters.

**Proposition 3** *Let $\mathcal{T}_1, \mathcal{T}_2 \in \mathbb{R}^{R \times R \times k_3 \times k_4}, \mathbf{X}_1, \mathbf{X}_2 \in \mathbb{R}^{k_1 \times R}, \mathbf{Y}_1, \mathbf{Y}_2 \in \mathbb{R}^{k_2 \times R}, R \leq \min(k_1, k_2)$ and the convolution kernel be $\mathcal{W} := (\mathcal{T}_1 \times_1 \mathbf{X}_1 \times_2 \mathbf{Y}_1) \odot (\mathcal{T}_2 \times_1 \mathbf{X}_2 \times_2 \mathbf{Y}_2)$. Then, the rank of the kernel satisfies $\mathrm{rank}(\mathcal{W}^{(1)}) = \mathrm{rank}(\mathcal{W}^{(2)}) \leq R^2$.*

*Proof.* According to Rabanser et al. (2017), the $1^{\mathrm{st}}$ and $2^{\mathrm{nd}}$ unfolding of tensors can be expressed as

$$\begin{aligned}
\mathcal{W}^{(1)} &= (\mathbf{X}_1 \mathcal{T}_1^{(1)} (\mathbf{I}^{(4)} \otimes \mathbf{I}^{(3)} \otimes \mathbf{Y}_1)^\top) \odot (\mathbf{X}_2 \mathcal{T}_2^{(1)} (\mathbf{I}^{(4)} \otimes \mathbf{I}^{(3)} \otimes \mathbf{Y}_2)^\top), \\
\mathcal{W}^{(2)} &= (\mathbf{Y}_1 \mathcal{T}_1^{(2)} (\mathbf{I}^{(4)} \otimes \mathbf{I}^{(3)} \otimes \mathbf{X}_1)^\top) \odot (\mathbf{Y}_2 \mathcal{T}_2^{(2)} (\mathbf{I}^{(4)} \otimes \mathbf{I}^{(3)} \otimes \mathbf{X}_2)^\top),
\end{aligned} \tag{5}$$

where $\mathbf{I}^{(3)} \in \mathbb{R}^{k_3 \times k_3}$ and $\mathbf{I}^{(4)} \in \mathbb{R}^{k_4 \times k_4}$ are identity matrices. Since $\mathcal{W}^{(1)}$ and $\mathcal{W}^{(2)}$ are matrices, we apply the same process used in Eq. 2, then we obtain $\mathrm{rank}(\mathcal{W}^{(1)}) = \mathrm{rank}(\mathcal{W}^{(2)}) \leq R^2$. $\qquad \square$

## A.2 ANALYSIS OF THE RANK PROPERTY

To demonstrate our propositions empirically, we sample the parameters randomly and count $\mathrm{rank}(\mathbf{W})$. When applying our parameterization to $\mathbf{W} \in \mathbb{R}^{100 \times 100}$, we set $r_{min} = 10$ by Corollary 1 to Proposition 2. We sample the entries of $\mathbf{X}_1, \mathbf{X}_2, \mathbf{Y}_1, \mathbf{Y}_2 \in \mathbb{R}^{100 \times 10}$ from the standard Gaussian distribution and repeat this experiment $1,000$ times.

As shown in Figure 6, we observe that our parameterization achieves the full rank with the probability of $100\%$ but requires 2.5 times fewer entries than the original $100 \times 100$ matrix. This empirical result demonstrates that our parameterization can span the full-rank matrix with fewer parameters efficiently.

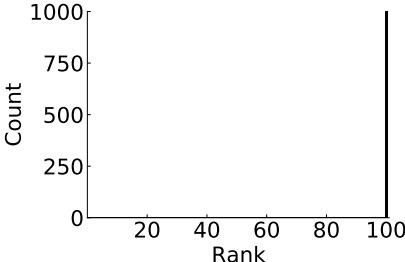

Figure 6: Histogram of $\mathrm{rank}(\mathbf{W})$. We apply `FedPara` to $\mathbf{W} \in \mathbb{R}^{100 \times 100}$ and set $r_1 = r_2 = 10$. We repeat this experiment $1,000$ times.

## B  ADDITIONAL TECHNIQUES

We devise additional techniques to improve the accuracy and stability of our parameterization. We consider injecting a non-linearity in `FedPara` and Jacobian correction regularization to further squeeze out the performance and stability. However, as will be discussed later, these are optional.

**Non-linear Function.**   The first is to apply a non-linear function before the Hadamard product. `FedPara` composites the weight as $\mathbf{W} = \mathbf{W}_1 \odot \mathbf{W}_2$, and we inject non-linearity as $\mathbf{W} = \sigma(\mathbf{W}_1) \odot \sigma(\mathbf{W}_2)$ as a practical design. This makes our algorithm departing from directly applying the proofs, *i.e.*, empirical heuristics. However, the favorable performance of our `FedPara` regardless of this non-linearity injection in practice suggests that our re-parameterization is a reasonable model to deploy.

The non-linear function $\sigma(\cdot)$ can be any non-linear function such as ReLU, Tanh, and Sigmoid, but ReLU and Sigmoid restrict the range to only positive, whereas Tanh has both negative and positive values of range $[-1, 1]$. The filters may learn to extract the distinct features, such as edge, color, and blob, which require both negative and positive values of the filters. Furthermore, the reasonably bounded range can prevent overshooting a large value by multiplication. Therefore, Tanh is suitable, and we use Tanh through experiments except for those of `pFedPara`.

**Jacobian Correction.**   The second is the Jacobian correction regularization, which induces that the Jacobians of $\mathbf{X}_1, \mathbf{X}_2, \mathbf{Y}_1$, and $\mathbf{Y}_2$ follow the Jacobian of the constructed matrix $\mathbf{W}$. Suppose that $\mathbf{X}_1, \mathbf{X}_2 \in \mathbb{R}^{m \times r}$ and $\mathbf{Y}_1, \mathbf{Y}_2 \in \mathbb{R}^{n \times r}$ are given. We construct the weight as $\mathbf{W} = \mathbf{W}_1 \odot \mathbf{W}_2 \in \mathbb{R}^{n \times r}$, where $\mathbf{W}_1 = \mathbf{X}_1 \mathbf{Y}_1^\top$ and $\mathbf{W}_2 = \mathbf{X}_2 \mathbf{Y}_2^\top$. Additionally, suppose that the Jacobian of $\mathbf{W}$ with respect to the objective function is given: $\mathbf{J}_\mathbf{W} = \frac{\partial \mathbf{L}}{\partial \mathbf{W}}$. We can compute the Jacobians of $\mathbf{X}_1, \mathbf{X}_2, \mathbf{Y}_1$, and $\mathbf{Y}_2$ with respect to the objective function and apply one-step optimization with SGD. For simplicity, we set the momentum and the weight decay to be zero. Then, we can compute the Jacobian of other variables using the chain rule:

$$\mathbf{J}_{\mathbf{W}_1} = \frac{\partial \mathbf{L}}{\partial \mathbf{W}_1} = \mathbf{J}_\mathbf{W} \odot \mathbf{W}_2, \quad \mathbf{J}_{\mathbf{X}_1} = \frac{\partial \mathbf{L}}{\partial \mathbf{X}_1} = \mathbf{J}_{\mathbf{W}_1} \mathbf{Y}_1^\top, \quad \mathbf{J}_{\mathbf{Y}_1} = \frac{\partial \mathbf{L}}{\partial \mathbf{Y}_1} = \mathbf{X}_1^\top \mathbf{J}_{\mathbf{W}_1},$$
$$\mathbf{J}_{\mathbf{W}_2} = \frac{\partial \mathbf{L}}{\partial \mathbf{W}_2} = \mathbf{J}_\mathbf{W} \odot \mathbf{W}_1, \quad \mathbf{J}_{\mathbf{X}_2} = \frac{\partial \mathbf{L}}{\partial \mathbf{X}_2} = \mathbf{J}_{\mathbf{W}_2} \mathbf{Y}_2^\top, \quad \mathbf{J}_{\mathbf{Y}_2} = \frac{\partial \mathbf{L}}{\partial \mathbf{Y}_2} = \mathbf{X}_2^\top \mathbf{J}_{\mathbf{W}_2}. \tag{6}$$

We update the parameters using SGD with the step size $\eta$ as follows:

$$\mathbf{X}_1' = \mathbf{X}_1 - \eta \mathbf{J}_{\mathbf{X}_1}, \quad \mathbf{Y}_1' = \mathbf{Y}_1 - \eta \mathbf{J}_{\mathbf{Y}_1},$$
$$\mathbf{X}_2' = \mathbf{X}_2 - \eta \mathbf{J}_{\mathbf{X}_2}, \quad \mathbf{Y}_2' = \mathbf{Y}_2 - \eta \mathbf{J}_{\mathbf{Y}_2}. \tag{7}$$

We can compute $\mathbf{W}'$, which is the constructed weight after one-step optimization:

$$\begin{aligned}
\mathbf{W}' &= (\mathbf{X}_1' \mathbf{Y}_1'^\top) \odot (\mathbf{X}_2' \mathbf{Y}_2'^\top) \\
&= \{(\mathbf{X}_1 - \eta \mathbf{J}_{\mathbf{X}_1})(\mathbf{Y}_1^\top - \eta \mathbf{J}_{\mathbf{Y}_1}^\top)\} \odot \{(\mathbf{X}_2 - \eta \mathbf{J}_{\mathbf{X}_2})(\mathbf{Y}_2^\top - \eta \mathbf{J}_{\mathbf{Y}_2}^\top)\} \\
&= \{\mathbf{X}_1 \mathbf{Y}_1^\top - \eta(\mathbf{J}_{\mathbf{X}_1} \mathbf{Y}_1^\top + \mathbf{X}_1 \mathbf{J}_{\mathbf{Y}_1}^\top) + \eta^2 \mathbf{J}_{\mathbf{X}_1} \mathbf{J}_{\mathbf{Y}_1}^\top\} \odot \{\mathbf{X}_2 \mathbf{Y}_2^\top - \eta(\mathbf{J}_{\mathbf{X}_2} \mathbf{Y}_2^\top + \mathbf{X}_2 \mathbf{J}_{\mathbf{Y}_2}^\top) + \eta^2 \mathbf{J}_{\mathbf{X}_2} \mathbf{J}_{\mathbf{Y}_2}^\top\} \\
&= (\mathbf{X}_1 \mathbf{Y}_1^\top) \odot (\mathbf{X}_2 \mathbf{Y}_2^\top) + \eta^4 (\mathbf{J}_{\mathbf{X}_1} \mathbf{J}_{\mathbf{Y}_1}^\top) \odot (\mathbf{J}_{\mathbf{X}_2} \mathbf{J}_{\mathbf{Y}_2}^\top) \\
&\quad - \eta^3 \{(\mathbf{J}_{\mathbf{X}_1} \mathbf{Y}_1^\top + \mathbf{X}_1 \mathbf{J}_{\mathbf{Y}_1}^\top) \odot (\mathbf{J}_{\mathbf{X}_2} \mathbf{J}_{\mathbf{Y}_2}^\top) + (\mathbf{J}_{\mathbf{X}_2} \mathbf{Y}_2^\top + \mathbf{X}_2 \mathbf{J}_{\mathbf{Y}_2}^\top) \odot (\mathbf{J}_{\mathbf{X}_1} \mathbf{J}_{\mathbf{Y}_1}^\top)\} \\
&\quad + \eta^2 \{(\mathbf{J}_{\mathbf{X}_1} \mathbf{Y}_1^\top + \mathbf{X}_1 \mathbf{J}_{\mathbf{Y}_1}^\top) \odot (\mathbf{J}_{\mathbf{X}_2} \mathbf{Y}_2^\top + \mathbf{X}_2 \mathbf{J}_{\mathbf{Y}_2}^\top) + (\mathbf{J}_{\mathbf{X}_2} \mathbf{J}_{\mathbf{Y}_2}^\top) \odot (\mathbf{X}_2 \mathbf{Y}_2^\top) + (\mathbf{J}_{\mathbf{X}_2} \mathbf{J}_{\mathbf{Y}_2}^\top) \odot (\mathbf{X}_1 \mathbf{Y}_1^\top)\} \\
&\quad - \eta \{(\mathbf{J}_{\mathbf{X}_1} \mathbf{Y}_1^\top + \mathbf{X}_1 \mathbf{J}_{\mathbf{Y}_1}^\top) \odot (\mathbf{X}_2 \mathbf{Y}_2^\top) + (\mathbf{J}_{\mathbf{X}_2} \mathbf{Y}_2^\top + \mathbf{X}_2 \mathbf{J}_{\mathbf{Y}_2}^\top) \odot (\mathbf{X}_1 \mathbf{Y}_1^\top)\}.
\end{aligned} \tag{8}$$

This shows that gradient descent and ascent are mixed, as shown in the signs of each term. We propose the Jacobian correction regularization to minimize the difference between $\mathbf{W}'$ and $\mathbf{W} - \eta \mathbf{J}_\mathbf{W}$, which induces our parameterization to follows the direction of $\mathbf{W} - \eta \mathbf{J}_\mathbf{W}$. The total objective function consists of the target loss function and the Jacobian correction regularization as:

$$\mathfrak{R} = L(\mathbf{X}_1, \mathbf{X}_2, \mathbf{Y}_1, \mathbf{Y}_2) + \frac{\lambda}{2} \|\mathbf{W}' - (\mathbf{W} - \eta \mathbf{J}_\mathbf{W})\|_2. \tag{9}$$

**Results.**   We evaluate the effects of each technique in Table 4. We train `VGG16` with group normalization on the CIFAR-10 IID setting during the same target rounds. We set $\gamma = 0.1$ and $\lambda = 10$. As

| Models | Accuracy |
|---|---|
| FedPara (base) | $82.45 \pm 0.35$ |
| + Tanh | $82.42 \pm 0.33$ |
| + Regularization | $82.38 \pm 0.30$ |
| + Both | $\mathbf{82.52} \pm 0.26$ |

Table 4: Accuracy of FedPara with additional techniques. 95% confidence intervals are presented with eight repetitions.

shown, the model with both Tanh and regularization has higher accuracy and lower variation than the base model. There is gain in accuracy and variance with both techniques, whereas only variance with only one technique.

Again, note that these additional techniques are not essential for FedPara to work; therefore, we can optionally use these techniques depending on the situation where the device has enough computing power.

## C DETAILS OF EXPERIMENT SETUP

In this section, we explain the details of the experiments, including datasets, models, and hyperparameters. We also summarize our FedPara and pFedPara into the pseudo algorithm. For implementation, we use PyTorch Distributed library (Paszke et al., 2019) and 8 NVIDIA GeForce RTX 3090 GPUs.

### C.1 DATASETS

**CIFAR-10.** CIFAR-10 (Krizhevsky et al., 2009) is the popular classification benchmark dataset. CIFAR-10 consists of $32 \times 32$ resolution images in 10 classes, with $6,000$ images per class. We use $50,000$ images for training and $10,000$ images for testing. For federated learning, we split training images into 100 partitions and assign one partition to each client. For the IID setting, we split the dataset into 100 partitions randomly. For the non-IID setting, we use the Dirichlet distribution and set the Dirichlet parameter as $0.5$ as suggested by He et al. (2020b;a).

**CIFAR-100.** CIFAR-100 (Krizhevsky et al., 2009) is the popular classification benchmark dataset. CIFAR-100 consists of $32 \times 32$ resolution images in 100 classes, with $6,000$ images per class. We use $50,000$ images for training and $10,000$ images for testing. For federated learning, we split training images into 50 partitions. For the IID setting, we split the dataset into 50 partitions randomly. For the non-IID setting, we use the Dirichlet distribution and set the Dirichlet parameter as $0.5$ as suggested by He et al. (2020b;a).

**CINIC-10.** CINIC-10 (Darlow et al., 2018) is a drop-in replacement for CIFAR-10 and also the popular classification benchmark dataset. CINIC-10 consists of $32 \times 32$ resolution images in 10 classes and three subsets: training, validation, and test. Each subset has $90,000$ images with $9,000$ per class, and we do not use the validation subset for training. For federated learning, we split training images into 100 partitions. For the IID setting, we split the dataset into 100 partitions randomly. For the non-IID setting, we use the Dirichlet distribution and set the Dirichlet parameter as $0.5$ as suggested by He et al. (2020b;a).

**MNIST.** MNIST (LeCun et al., 1998) is a popular handwritten number image dataset. MNIST consists of $70,000$ number of $28 \times 28$ resolution images in 10 classes. We use $60,000$ images are for training and $10,000$ images for testing. We do not use MNIST IID-setting, and we split the dataset so that clients have at most two classes as suggested by McMahan et al. (2017) for a highly-skew non-IID setting.

**FEMNIST.** FEMNIST (Caldas et al., 2018) is a handwritten image dataset for federated settings. FEMNIST has 62 classes and $3,550$ clients, and each client has $226.83$ data samples on average of $28 \times 28$ resolution images. FEMNIST is the non-IID dataset labeled by writers.

**Shakespeare.** Shakespeare (Shakespeare, 1994) is a next word prediction dataset for federated learning settings. Shakespeare has 80 classes and $1,129$ clients, and each client has $3,743.2$ data samples on average (Caldas et al., 2018).

## C.2 MODELS

**VGG16.** PyTorch library (Paszke et al., 2019) provides `VGG16` with batch normalization, but we replace the batch normalization layers with the group normalization layers as suggested by Hsieh et al. (2020) for federated learning. We also modify the FC layers to comply with the number of classes. The dimensions of the output features in the last three FC layers are $512$–$512$–$\langle \#\text{classes}\rangle$, sequentially. We do not apply our parameterization to the last three FC layers and set the same $\gamma$ to all convolutional layers in the model for simplicity. For reference purpose, Table 5 shows the number of parameters corresponding to each $\gamma$.

| $\gamma$ | No. parameters | |
|---|---|---|
| | 10-classes | 100-classes |
| original | 15.25M | 15.30M |
| 0.1 | 1.55 M | 1.59 M |
| 0.2 | 2.33 M | 2.38 M |
| 0.3 | 3.31 M | 3.36 M |
| 0.4 | 4.45 M | 4.50 M |
| 0.5 | 5.79 M | 5.84 M |
| 0.6 | 7.33 M | 7.38 M |
| 0.7 | 9.01 M | 9.05 M |
| 0.8 | 10.90 M | 10.94 M |
| 0.9 | 12.92 M | 12.96 M |

Table 5: $\gamma$'s and their corresponding numbers of parameters for `VGG16`$_{\text{ori.}}$ and `VGG16`$_{\text{FedPara}}$.

**Two FC Layers.** In personalization experiments, we use two FC layers as suggested by McMahan et al. (2017) but modify the size of the hidden features corresponding to the number of classes in the datasets. The dimensions of the output features in two FC layers are $256$ and the number of classes, respectively; *i.e.*, $256$–$\langle\#\text{classes}\rangle$. We do not use other layers, such as normalization and dropout, and set $\gamma = 0.5$ for `pFedPara`.

**LSTM.** For the Shakespeare dataset, we use two-layer LSTM as suggested by McMahan et al. (2017) and Acar et al. (2021). We set the hidden dimension as $256$ and the number of classes as 80. We also apply the weight normalization technique on original parameterization, low-rank parameterization, and `FedPara`.

## C.3 FEDPARA & PFEDPARA

---
**Algorithm 1:** `FedPara`

---
**Input:** rounds $T$, parameters $\{\mathbf{X}_{1l}, \mathbf{X}_{2l}, \mathbf{Y}_{1l}, \mathbf{Y}_{2l}\}_{l=1}^{l=L}$ where $\{\mathbf{X}_{1l}, \mathbf{X}_{2l}, \mathbf{Y}_{1l}, \mathbf{Y}_{2l}\}$ is the $l^{th}$ layer of the model and $L$ is the number of layers

**for** $t = 1, 2, \ldots, T$ **do**
    Sample the subset $S$ of clients;
    **for** *each client* $c \in S$ **do**
        Download $\{\mathbf{X}_{1l}, \mathbf{X}_{2l}, \mathbf{Y}_{1l}, \mathbf{Y}_{2l}\}_{l=1}^{l=L}$ from the server;
        Optimize($\{\mathbf{X}_{1l}, \mathbf{X}_{2l}, \mathbf{Y}_{1l}, \mathbf{Y}_{2l}\}_{l=1}^{l=L}$);
        Upload $\{\mathbf{X}_{1l}, \mathbf{X}_{2l}, \mathbf{Y}_{1l}, \mathbf{Y}_{2l}\}_{l=1}^{l=L}$ to the server;
    **end**
    Aggregate $\{\mathbf{X}_{1l}, \mathbf{X}_{2l}, \mathbf{Y}_{1l}, \mathbf{Y}_{2l}\}_{l=1}^{l=L}$;
**end**

---

---

**Algorithm 2:** `pFedPara`

---

**Input:** rounds $T$, parameters $\{\mathbf{X}_{1l}, \mathbf{X}_{2l}, \mathbf{Y}_{1l}, \mathbf{Y}_{2l}\}_{l=1}^{l=L}$ where $\{\mathbf{X}_{1l}, \mathbf{X}_{2l}, \mathbf{Y}_{1l}, \mathbf{Y}_{2l}\}$ is the $l^{th}$ layer of the model and $L$ is the number of layers

Transmit $\{\mathbf{X}_{1l}, \mathbf{X}_{2l}, \mathbf{Y}_{1l}, \mathbf{Y}_{2l}\}_{l=1}^{l=L}$ to clients to train the same initial point at start;

**for** $t = 1, 2, \ldots, T$ **do**

    Sample the subset $S$ of clients;

    **for** *each client* $c \in S$ **do**

        Download half of parameters $\{\mathbf{X}_{1l}, \mathbf{Y}_{1l}\}_{l=1}^{l=L}$ from the server;

        Optimize($\{\mathbf{X}_{1l}, \mathbf{X}_{2l}, \mathbf{Y}_{1l}, \mathbf{Y}_{2l}\}_{l=1}^{l=L}$);

        Upload $\{\mathbf{X}_{1l}, \mathbf{Y}_{1l}\}_{l=1}^{l=L}$ to the server;

    **end**

    Aggregate $\{\mathbf{X}_{1l}, \mathbf{Y}_{1l}\}_{l=1}^{l=L}$;

**end**

---

We summarize our two methods, `FedPara` and `pFedPara`, into Algorithms 1 and 2 for apparent comparison. In these algorithms, we mainly use the popular and standard algorithm, `FedAvg`, as a backbone optimizer, but we can switch with other optimizer and aggregate methods. As mentioned in Section B, we can consider the additional techniques. In the `FedPara` experiments, we use the regularization to Algorithm 1, and set the regularization coefficient $\lambda$ as 1.0. In `pFedPara` experiments, we do not apply the additional techniques to Algorithm 2.

## C.4 HYPER-PARAMETERS OF BACKBONE OPTIMIZER

`FedAvg` (McMahan et al., 2017) is the most popular algorithm in federated learning. The server samples $S$ number of clients as a subset in each round, each client of the subset trains the model locally by $E$ number of SGD epochs, and the server aggregates the locally updated models and repeats these processes during the total rounds $T$. We use `FedAvg` as a backbone optimization algorithm, and its hyper-parameters of our experiments, such as the initial learning rate $\eta$, local batch size $B$, and learning rate decay $\tau$, are described in Table 6.

| Models | CIFAR-10 | | CIFAR-100 | | CINIC-10 | | LSTM | | FEMNIST & MNIST |
|---|---|---|---|---|---|---|---|---|---|
| | IID | non-IID | IID | non-IID | IID | non-IID | IID | non-IID | |
| $K$ | 16 | 16 | 8 | 8 | 16 | 16 | 16 | 16 | 10 |
| $T$ | 200 | 200 | 400 | 400 | 300 | 300 | 500 | 500 | 100 |
| $E$ | 10 | 5 | 10 | 5 | 10 | 5 | 1 | 1 | 5 |
| $B$ | 64 | 64 | 64 | 64 | 64 | 64 | 64 | 64 | 10 |
| $\eta$ | 0.1 | 0.1 | 0.1 | 0.1 | 0.1 | 0.1 | 1.0 | 1.0 | 0.1-0.01 |
| $\tau$ | 0.992 | 0.992 | 0.992 | 0.992 | 0.992 | 0.992 | 0.992 | 0.992 | 0.999 |
| $\lambda$ | 1 | 1 | 1 | 1 | 1 | 1 | 0 | 0 | 0 |

Table 6: Hyper-parameters of our `FedPara` with `FedAvg`

## C.5 HYPER-PARAMETERS OF OTHER OPTIMIZERS

For compatibility experiment, we combine `FedPara` with other optimization-based FL algorithms: `FedProx` (Li et al., 2020), `SCAFFOLD` (Karimireddy et al., 2020), `FedDyn` (Acar et al., 2021), and `FedAdam` (Reddi et al., 2021). `FedProx` (Li et al., 2020) imposes a proximal term to the objective function to mitigate heterogeneity; `SCAFFOLD` (Karimireddy et al., 2020) allows clients to reduce the variance of gradients by introducing auxiliary variables; `FedDyn` (Acar et al., 2021) introduces dynamic regularization to reduce the inconsistency between minima of the local device level empirical losses and the global one; `FedAdam` employs Adam (Kingma & Ba, 2015) at the server-side instead of the simple model average.

They need a local optimizer to update the model in each client, and we use the SGD optimizer for a fair comparison, and the SGD configuration is the same as that of `FedAvg`. The four algorithms have additional hyper-parameters. `FedProx` has a proximal coefficient $\mu$, and we set $\mu$ as 0.1. SCAFFOLD has Options I and II to update the control variate, and we use Option II with global learning rate $\eta_g$ (= 1.0). `FedDyn` has the hyper-parameter $\alpha$ (= 0.1) in the regularization. `FedAdam` uses Adam optimizer to aggregate the updated models at the server-side, and we use the

| Network speed | Model | $t_{comp.}$ | $t_{comm.}$ | $t$ |
|---|---|---|---|---|
| 2 Mbps | $\text{VGG16}_{\text{ori.}}$ | 1.64 sec. | 470.2 sec. | 471.84 sec. |
| | $\text{VGG16}_{\texttt{FedPara}}$ ($\gamma$=0.1) | 2.34 sec. | 47.2 sec. | 49.54 sec. ($\times$ **9.52**) |
| 10 Mbps | $\text{VGG16}_{\text{ori.}}$ | 1.64 sec. | 94.04 sec. | 94.68 sec. |
| | $\text{VGG16}_{\texttt{FedPara}}$ ($\gamma$=0.1) | 2.34 sec. | 9.44 sec. | 11.78 sec. ($\times$ **8.04**) |
| 50 Mbps | $\text{VGG16}_{\text{ori.}}$ | 1.64 sec. | 18.61 sec. | 20.25 sec. |
| | $\text{VGG16}_{\texttt{FedPara}}$ ($\gamma$=0.1) | 2.34 sec. | 1.88 sec. | 4.22 sec. ($\times$ **4.80**) |

Table 7: The required time during one round. We denote the computation time, the communication time, and the total time during one round as $t_{comp.}$, $t_{comm.}$, and $t$, respectively. We set the network speeds as 2, 10, and 50 Mbps.

| Network speed | Model | Training time |
|---|---|---|
| 2 Mbps | $\text{VGG16}_{\text{ori.}}$ | 880.77 min. |
| | $\text{VGG16}_{\texttt{FedPara}}$ ($\gamma$=0.1) | 94.95 min. ($\times$ **9.28**) |
| 10 Mbps | $\text{VGG16}_{\text{ori.}}$ | 176.74 min. |
| | $\text{VGG16}_{\texttt{FedPara}}$ ($\gamma$=0.1) | 22.58 min. ($\times$ **7.83**) |
| 50 Mbps | $\text{VGG16}_{\text{ori.}}$ | 37.80 min. |
| | $\text{VGG16}_{\texttt{FedPara}}$ ($\gamma$=0.1) | 8.09 min. ($\times$ **4.67**) |

Table 8: The real training time to achieve the target accuracy. We set the network speeds as 2, 10, and 50 Mbps, and the required rounds for $\text{VGG16}_{\text{ori.}}$ is 112, $\texttt{FedPara}$ ($\gamma = 0.1$) is 115 to achieve the same target accuracy in the CIFAR-10 IID setting.

parameters $\beta_1 = 0.9$, $\beta_2 = 0.99$, the global learning rate $\eta_g = 0.01$, and the local learning rate $\eta = 10^{-1.5}$ for Adam optimizer.

# D  ADDITIONAL EXPERIMENTS

In this section, we simulate the computation time, the communication time, and total training time in Section D.1, experiment about other models in Section D.2, and compare our method and the quantization approach in Section D.3.

## D.1  TRAINING TIME

We can compute the elapsed time during one round in FL by $t = t_{comp.} + t_{comm.}$, where $t_{comp.}$ is the computation time of training the model on local data for several epochs and $t_{comm.}$ is the communication time of downloading and uploading the updated model. We estimate the computation time by measuring the elapsed time during local epochs. We compute the communication time by $\frac{2 \cdot \text{model size (Mbyte)}}{\text{network speed (Mbyte/s)}}$, considering upload and download of the model in one round.

Since it is challenging to experiment in a real heterogeneous network environment, we follow the simple standard network simulation setting widely used in the communication literature (Zhu et al., 2020; Jeon et al., 2020). They assume the homogeneous link quality by the average quality to simplify complex network environments in FL communication simulation, i.e., the network speeds are identical for all clients.

We compare the elapsed time per round of $\text{VGG16}_{\text{ori.}}$ and $\text{VGG16}_{\texttt{FedPara}}$ on different network speeds such as 2, 10, and 50 Mbps. As shown in Table 7, the communication time is larger than the computation time indicating that the communication is a bottleneck in FL as expected. Although $\text{VGG16}_{\texttt{FedPara}}$ takes more computation time than $\text{VGG16}_{\text{ori.}}$ due to the weight composition time, $\text{VGG16}_{\texttt{FedPara}}$ decreases the communication time by about ten times and the total time by 4.80 to 9.52 times. Table 8 shows the total training time to achieve the same accuracy on the CIFAR-10 IID setting. Although $\texttt{FedPara}$ needs three rounds more than the original parameterization, $\text{VGG16}_{\texttt{FedPara}}$ requires 4.67 to 9.68 times less training time than $\text{VGG16}_{\text{ori.}}$ because of communication efficient parameterization.

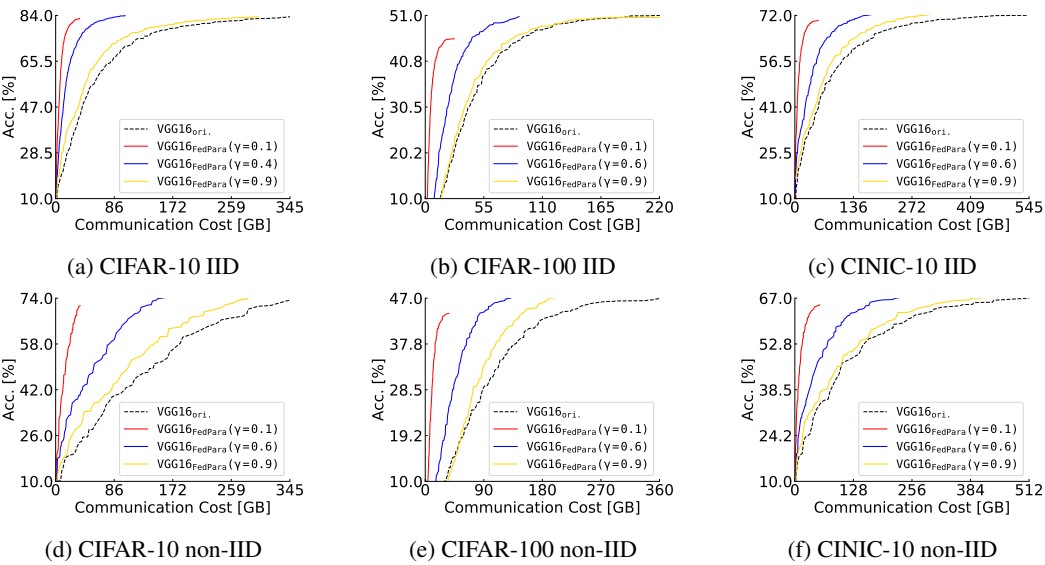

Figure 7: (a-f) Accuracy [%] ($y$-axis) vs. communication costs [GBytes] ($x$-axis) of $\text{VGG16}_{\text{ori.}}$ and $\text{VGG16}_{\text{FedPara}}$. Broken black line represents $\text{VGG16}_{\text{ori.}}$, red solid line $\text{VGG16}_{\text{FedPara}}$ with low $\gamma$, blue solid line $\text{VGG16}_{\text{FedPara}}$ with mid $\gamma$, and yellow solid line $\text{VGG16}_{\text{FedPara}}$ with high $\gamma$

| Model | Acc. of 200 rounds | Acc. of 1000 rounds (gain) |
|---|---|---|
| Original | 83.68 | 84.1 (+ 0.42) |
| FedPara($\gamma = 0.1$) | 82.88 | 83.35 (+ 0.47) |
| FedPara($\gamma = 0.2$) | 82.53 | 83.34 (+ 0.81) |
| FedPara($\gamma = 0.3$) | 83.11 | 83.94 (+ 0.83) |
| FedPara($\gamma = 0.4$) | 84.05 | 84.59 (+ 0.54) |
| FedPara($\gamma = 0.5$) | 83.82 | 84.57 (+ 0.75) |
| FedPara($\gamma = 0.6$) | 83.63 | 84.16 (+ 0.53) |
| FedPara($\gamma = 0.7$) | 83.79 | 84.24 (+ 0.45) |
| FedPara($\gamma = 0.8$) | 83.40 | 84.00 (+ 0.6) |

Table 9: The accuracy of $\text{VGG16}$ with original and our parameterization on the CIFAR-10 IID setting during 200 and 1000 rounds.

### D.2 OTHER MODELS

**VGG16** As shown in Figure 7, we compare $\text{VGG16}_{\text{ori.}}$ and $\text{VGG16}_{\text{FedPara}}$ with three different $\gamma$ values. Note that a higher $\gamma$ uses more parameters than a lower $\gamma$. Compared to $\text{VGG16}_{\text{ori.}}$, $\text{VGG16}_{\text{FedPara}}$ achieves comparable accuracy and even higher accuracy with a high $\gamma$. $\text{VGG16}_{\text{FedPara}}$ also requires fewer communication costs than $\text{VGG16}_{\text{ori.}}$.

We also investigate how much accuracy is increased for longer rounds. Table 9 shows that the accuracy is increased in the long round experiment, but the tendency of the 1000 round training is consistent with the 200 round training.

**ResNet18** We demonstrate the consistent effectiveness of our method with another architecture, $\text{ResNet18}$. For $\text{ResNet18}$, we train without replacing batch normalization layers, set $\gamma$ of the first layer, the second layer, and the $1 \times 1$ convolution layers as 1.0, and adjust $\gamma$ of remaining layers to control the $\text{ResNet18}_{\text{FedPara}}$ size; $\gamma = 0.1$ for small model size, $\gamma = 0.6$ for mid model size, and $\gamma = 0.9$ for large model size. To train the $\text{ResNet18}_{\text{FedPara}}$ in FL, we set the batch size to 10.

As revealed in Figure 8a, $\text{ResNet18}_{\text{FedPara}}$ has comparable accuracy and uses fewer communication costs than $\text{ResNet18}_{\text{ori.}}$, of which the results are consistent with the $\text{VGG16}$ experiments. Figure 8b shows the communication costs required for model training to achieve the same tar-

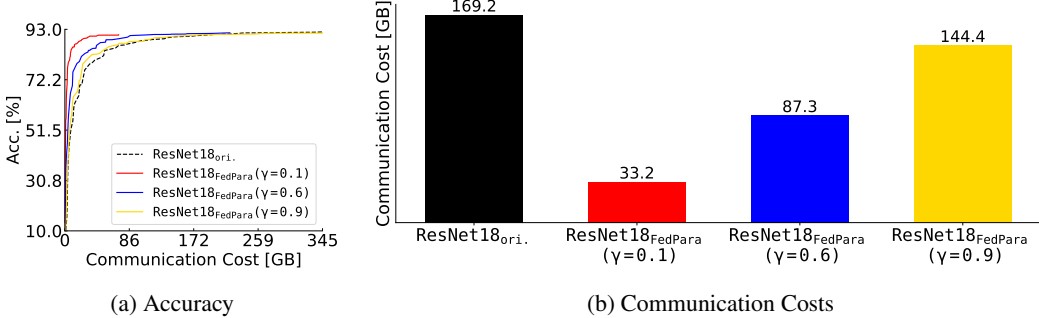

(a) Accuracy

(b) Communication Costs

Figure 8: (a) Accuracy [%] (y-axis) vs. communication costs [GBytes] ($x$-axis) of $\mathtt{ResNet18_{ori.}}$ and $\mathtt{ResNet18_{FedPara}}$ with three $\gamma$ values. (b) Size comparison of transferred parameters, which is expressed as communication costs [GBytes] ($y$-axis), for the same target accuracy 90%. (a, b): Black represents $\mathtt{ResNet18_{ori.}}$, and red, blue, and yellow represents $\mathtt{ResNet18_{FedPara}}$ with low, mid, and high $\gamma$, respectively.

| Model | Acc. | # parameters (ratio) |
|---|---|---|
| $\mathtt{VGG16_{Pufferfish}}$ | 82.07 | 0.33 |
| $\mathtt{VGG16_{Pufferfish}}$ | 82.42 | 0.44 |
| $\mathtt{VGG16_{FedPara}}$ ($\gamma = 0.2$) | 82.53 | 0.15 |
| $\mathtt{VGG16_{FedPara}}$ ($\gamma = 0.4$) | 84.05 | 0.29 |

Table 10: The accuracy of $\mathtt{VGG16_{Pufferfish}}$ and $\mathtt{VGG16_{FedPara}}$ on CIFAR-10 IID setting. # parameters is the the ratio of each model parameter when the number of parameters of $\mathtt{VGG16_{ori.}}$ is set 1.0.

get accuracy. Our $\mathtt{ResNet18_{FedPara}}$ needs 1.17 to 5.1 times fewer communication costs than $\mathtt{ResNet18_{ori.}}$, and the results demonstrate that $\mathtt{FedPara}$ is also applicable to the $\mathtt{ResNet}$ structure.

**Pufferfish** Pufferfish (Wang et al., 2021) is similar to our method, where they use partially pre-factorized networks and employ the hybrid architecture by maintaining original size weights to minimize the accuracy loss due to the low-rank constraints. Since this work can be directly applied in the FL setup, we compare the parameterization of PufferFish and FedPara as follows.

We train VGG16 with PufferFish and FedPara on the CIFAR-10 IID dataset in FL, and evaluate the models according to varying the number of parameters. As shown in Table 10, our FedPara has higher accuracy with fewer parameters compared with PufferFish. Although PufferFish is superior to the naive low-rank pre-decomposition method due to its hybrid architecture, we think the hybrid architecture of PufferFish still suffers from the low-rank constraints on the top layers. However, our method is free from such limitations both theoretically and empirically.

**LSTM** We train $\mathtt{LSTM_{ori.}}$, $\mathtt{LSTM_{low}}$, and $\mathtt{LSTM_{FedPara}}$ on the Shakespeare dataset. The parameters ratio of $\mathtt{LSTM_{low}}$ and $\mathtt{LSTM_{FedPara}}$ are about 16% and 19% of the $\mathtt{LSTM_{ori.}}$ parameters, respectively. Table 11 shows that $\mathtt{LSTM_{FedPara}}$ outperforms $\mathtt{LSTM_{low}}$ and $\mathtt{LSTM_{ori.}}$ on the IID setting. In the non-IID setting, $\mathtt{LSTM_{FedPara}}$ accuracy is higher than $\mathtt{LSTM_{low}}$ and slightly lower than $\mathtt{LSTM_{ori.}}$ with only 19% of parameters. Therefore, our parameterization can be applied to general neural networks.

### D.3 QUANTIZATION

$\mathtt{FedPAQ}$ (Reisizadeh et al., 2020) is a quantization approach to reduce communication costs for FL. To compare the accuracy and transferred size per round, we train VGG16 on the CIFAR-10 IID setting. We consider both downlink and uplink to evaluate communication costs. We quantize the model from 32 bits floating-point numbers to 16 bits floating-point numbers.

| Model | Acc. (IID) | Acc. (non-IID) | # parameters (ratio) |
|---|---|---|---|
| $\text{LSTM}_{\text{ori.}}$ | 60.17 | 52.66 | 1.0 |
| $\text{LSTM}_{\text{low}}$ | 54.59 | 51.24 | 0.16 |
| $\text{LSTM}_{\texttt{FedPara}}$ ($\gamma = 0.0$) | 63.65 | 51.56 | 0.19 |

Table 11: The accuracy of LSTMs on IID and non-IID setting. # parameters is the the ratio of each model parameter when the number of parameters of $\text{LSTM}_{\text{ori.}}$ is set 1.0.

| Model | Acc. | Transferred size per round |
|---|---|---|
| FedAvg | 83.68 | 122MB |
| FedPAQ | 82.67 | 91.4MB |
| FedPara ($\gamma = 0.5$) | **83.82** | 46.4MB |
| + FedPAQ | 83.58 | **34.74MB** |

Table 12: The accuracy of VGG16 on CIFAR-10 IID setting for original, quantization, and our parameterization models.

Table 12 shows the comparison between FedPara and FedPAQ as well as those integration. Compared to FedPAQ, FedPara transfers 1.96 times lower bits per round because FedPAQ only reduces the uplink communication cost. Compared with FedAvg, FedPara achieves 0.14% higher accuracy, but FedPAQ 1.01% lower accuracy. FedPara can transfer the full information of updated weights to the server, whereas FedPAQ loses the updated weight information due to the quantization. Furthermore, since the way FedPara reduces communication costs is different from quantization, we integrate FedPara with FedPAQ as an extension. Combining our method with FedPAQ reduces communication costs by 25% further from our FedPara without the integration while having a minor accuracy drop, 0.1%, from that of FedAvg.

