# OpenReview forum: "FedPara: Low-rank Hadamard Product for Communication-Efficient Federated Learning"
_ICLR.cc/2022/Conference — ICLR 2022 Poster_

### Official Review · Reviewer_M7KC · 2021-10-28

**Correctness:** 4
**Technical Novelty And Significance:** 3
**Empirical Novelty And Significance:** 3
**Recommendation:** 6
**Confidence:** 2

**Main Review:**

I am not very familiar with reparametrization, and I am not able to comment on the significance of this paper.

I think that the idea of this paper is interesting, using the Hadamard product of two low-rank matrices may only increase the model size by a factor of 2, but may drastically increase the expressiveness of the parametrization.

As for the experiments, I think the experiment design is generally enough. The experiment compares FedPara with original training methods with no reparametrization, and low-rank reparametrization without the Hadamard product on different machine learning models, e.g. VGG and LSTM.

Below are some suggestions and questions:
1. In the experiment, the authors compare FedPara with original low-rank reparametrization and without reparametrization, and the optimization algorithms are chosen to be the same, e.g. FedAvg, SCAFFOLD, etc. I wonder what are the results between FedPara and gradient compression algorithms, for example, quantization. Or if the result can become better if using gradient compression schemes in FedPara instead of local steps. It may also be good to add some experiment results with compression.

2. FedPara converges much faster than the original models in terms of the communication bits. How about the final accuracy? What is the tradeoff between communication efficiency and the testing accuracy?

3. Although the low-rank constraints reduce the number of communication bits during the training procedures, it may increase the local computational cost since the gradient of the new model may become more complicated. In your experiment, what is the local training cost, e.g. the local training time between two communications, or the number of flops for one backpropagation?

**Summary Of The Paper:**

The paper introduces FedPara, which is a low-rank parametrization method for the neural networks aiming to reduce the total number of communication bits while preserving the model accuracy in the federated learning scenario. Besides FedPara, which is designed for federated learning applications, the paper also generalizes FedPara into pFedPara, which is designed for personalized federated learning.

The novelty of this paper includes using the Hadamard product in the low-rank approximation. Compared with directly using low-rank approximation, using Hadamard produce can increase the expressiveness of the parametrization.

The paper also includes empirical results to show the effectiveness of FedPara/pFedPara in the federated learning setting.

**Summary Of The Review:**

I am not very familiar with reparametrization and its corresponding related works, and I am not able to comment on the significance. However, I think that the technique used in this paper is interesting and simple, which can be applied to real applications. The experiments shown in the paper are enough. Some questions and small suggestions are listed in the main review part.

---

> ### Author Response · Authors · 2021-11-13
> **Response to Reviewer M7KC's Review**
>
> We thank the reviewer for the time and comments. We address the concerns and the questions below:
>
> **Q1. I wonder what are the results between FedPara and gradient compression algorithms, for example, quantization. Or if the result can become better if using gradient compression schemes in FedPara instead of local steps. It may also be good to add some experiment results with compression.**
>
> FedPAQ [D1] has been proposed to apply quantization when uploading the model from clients to the server in FL. To compare quantization and FedPara, we train VGG16 on the CIFAR-10 IID setting. We quantize 32 bits floating-point to 16 bits floating points for FedPAQ. The below table shows the accuracy and transferred size per round:
>
>
>     Model              Acc.     Transferred size per round
>     ======================================================
>     FedAvg             83.68    122 MB
>     FedPAQ             82.67    91.4 MB
>     ------------------------------------------------------
>     FedPara (γ=0.5)    83.82    46.4 MB
>     + FedPAQ           83.58    34.74 MB
>     ======================================================
>
> In terms of communication costs, FedPara transfers 1.96 times lower bits round than FedPAQ. This is because FedPAQ only reduces the upload model size. Combining our method with FedPAQ reduces communication costs by 25% further from our FedPara while having a minor accuracy drop, 0.1%, from that of FedAvg.
>
> We have added these additional results in Section D.3 of the supplementary material (see pages 22-23).
>
> [D1] Reisizadeh, Amirhossein and Mokhtari, Aryan and Hassani, Hamed and Jadbabaie, Ali and Pedarsani, Ramtin, Fedpaq: A communication-efficient federated learning method with periodic averaging and quantization, AISTATS, 2020, http://proceedings.mlr.press/v108/reisizadeh20a/reisizadeh20a.pdf
>
> **Q2. FedPara converges much faster than the original models in terms of the communication bits. How about the final accuracy? What is the tradeoff between communication efficiency and the testing accuracy?**
>
> The below table shows the accuracy of the CIFAR-10 IID setting during 1000 rounds:
>
>     ====================================================================
>         γ       Acc. of 200 rounds    Acc. of 1000 rounds [difference]
>     --------------------------------------------------------------------
>       0.1             82.88                    83.35 [+ 0.47]
>       0.2             82.53                    83.34 [+ 0.81]
>       0.3             83.11                    83.94 [+ 0.83]
>       0.4             84.05                    84.59 [+ 0.54]
>       0.5             83.82                    84.57 [+ 0.75]
>       0.6             83.63                    84.16 [+ 0.53]
>       0.7             83.79                    84.24 [+ 0.45]
>       0.8             83.40                    84.00 [+ 0.6]
>     --------------------------------------------------------------------
>      original         83.68                    84.10 [+ 0.42]
>     ====================================================================
>
> We train the models for 1000 rounds, but we observe that the accuracy is saturated before 420 rounds. The tendency of the 1000 rounds training is consistent with the 200 rounds training, so the tradeoff between communication efficiency and testing accuracy is consistent.
>
> We have reflected the results in Table 9 and Section D.2 of the supplementary material (See page 21).

---

> > ### Author Response · Authors · 2021-11-15
> > **Response to Reviewer M7KC's Review**
> >
> > **Q3. In your experiment, what is the local training cost, e.g. the local training time between two communications, or the number of flops for one backpropagation?**
> >
> > In the initial submission, we reported the computation, communication, and training time in Tables 7 and 8, where the computation and training times other than the communication time correspond to the local training times.
> >
> > We simulated the homogeneous communication link quality, which is a widely used setting in the communication literature. We compare the elapsed time on different network speeds such as 2, 10, and 50 Mbps for FL. Table 7 is the training time for one round and is shown as below:
> >
> >     ===================================================================================
> >     Network speed    Model        Computation time    Communication time     Training time
> >     -----------------------------------------------------------------------------------
> >       2 Mbps        Original          1.64 sec.            470.2 sec.         471.84 sec.
> >       2 Mbps        FedPara (γ=0.1)   2.34 sec.            47.2 sec.          49.54 sec.
> >     -----------------------------------------------------------------------------------
> >       5 Mbps        Original          1.64 sec.            94.04 sec.         94.68 sec.
> >       5 Mbps        FedPara (γ=0.1)   2.34 sec.            9.44 sec.          11.78 sec.
> >     -----------------------------------------------------------------------------------
> >       10 Mbps        Original         1.64 sec.            18.61 sec.         20.25 sec.
> >       10 Mbps        FedPara (γ=0.1)  2.34 sec.            1.88 sec.          4.22 sec.
> >     ===================================================================================
> >
> > For one round, the computation time of our and original parameterization is 2.34 and 1.64 sec., respectively. The computation time of FedPara is slower than the original one because of the construction of weights.
> >
> > The communication time proportional to the transferred bits is much lower than the original. We assume the homogeneous communication quality, so the elapsed time between two communications is the same as the above table. As a result, FedPara requires 4.80 to 9.52 times faster training time per round than the original.
> >
> > The detailed discussion of the simulation results can be found in the supplementary material (See Section D.1 on page 20).

---

> > > ### Comment · Reviewer_M7KC · 2021-12-06
> > > **After Rebuttal**
> > >
> > > Dear Authors,
> > >
> > > I read the rebuttal a long time ago and forgot to reply here. My concerns are alleviated, but due to my lack of background in parametrization literature, I remain the score 6 and told AC my concerns are addressed.

---

> > > > ### Author Response · Authors · 2021-12-06
> > > > **Response to Reviewer M7KC's response**
> > > >
> > > > Thank the reviewer for the comments and suggestions. We believe that the suggestions have improved our submission. If the reviewer has any remaining feedback, we would be happy to discuss it further.

---

### Official Review · Reviewer_3Rte · 2021-11-01

**Correctness:** 3
**Technical Novelty And Significance:** 2
**Empirical Novelty And Significance:** 3
**Recommendation:** 8
**Confidence:** 2

**Main Review:**

The paper proposes a low-rank parameterization method to help improve the communication efficiency of federated learning. The work provides theoretical justification of the effectiveness in terms of the maximal rank the parameterization can express. The work shows extensive empirical studies of the proposed method on multiple FL datasets. The results indicate the superiority of the proposed method in terms of model accuracy and communication cost.

Questions and comments:
1. In terms of expressiveness, what is the key limitation of the Hadamard product parameterization compared with general unconstrained parameterization?
2. The paper does not discuss any related work about the Hadamard product parameterization. Is this a new idea? Or what is the difference compared to other literature involving Hadamard product parameterization?
3. In terms of computational cost, how does training the Hadamard product parameterization compare to previous low-rank approaches?

**Summary Of The Paper:**

The paper proposes a communication-efficient parameterization methodology for federated learning tasks through the Hadamard product of low-rank weights. Such parameterization has better model expressiveness in terms of minimal parameters to achieve a maximal rank, compared with conventional low-rank approaches. Using such parameterization, the work designs the corresponding communication-efficient federated learning framework which only uploads a low-rank part of parameters of each level to the server. The work presents empirical studies of the proposed algorithm and validates its effectiveness in terms of accuracy vs communication costs.

**Summary Of The Review:**

The work proposes an effective low-rank parameterization that shows the superiority of communication efficiency in federated learning tasks. The idea is relatively interesting to the reviewer. The overall presentation of the idea and empirical results is clear.

---

> ### Author Response · Authors · 2021-11-13
> **Response to Reviewer 3Rte's Review**
>
> We thank the reviewer for the time and constructive comments. We discuss the concerns and the questions below:
>
> **Q1. In terms of expressiveness, what is the key limitation of the Hadamard product parameterization compared with general unconstrained parameterization?**
>
> As mentioned in the first paragraph of the discussion section, the Hadamard product parameterization requires several multiplications to construct weights during training. Several multiplications may cause numerically unstable issues (underflow or overflow), so optimization might be complicated due to numerical instability.
>
> Several multiplications also may introduce preferred statistical distributions of weights and features. Since the statistical analysis has been used to study weight initialization and normalization, it is interesting to research our parameterization’s weights and features distribution in the future. The view of statistical distribution might give how to initialize and optimize the weight better when using our parameterization.
>
> We have added the above discussion in Section 5 (See the discussion section on page 9).
>
> **Q2. The paper does not discuss any related work about the Hadamard product parameterization. Is this a new idea? Or what is the difference compared to other literature involving Hadamard product parameterization?**
>
> To our best knowledge, it is the first time to use the Hadamard product of low-rank matrices as parameterization for lightweight modeling. While quite different, the closest work we found is low-rank bilinear pooling [C1]. They use the Hadamard product and low-rank matrices to reduce computational costs, but the model is different and works on activation, not weight.
>
> We would much appreciate it if any related work is recommended to cite and to discuss further.
>
> [C1] Kim, Jin-Hwa and On, Kyoung-Woon and Lim, Woosang and Kim, Jeonghee and Ha, Jung-Woo and Zhang, Byoung-Tak, Hadamard product for low-rank bilinear pooling, ICLR, 2016, https://openreview.net/forum?id=r1rhWnZkg
>
> **Q3. In terms of computational cost, how does training the Hadamard product parameterization compare to previous low-rank approaches?**
>
> As shown in Tables 7 and 8 on page 20 and mentioned in the discussion section, FedPara has a little higher computation cost than the original one because of constructing the weights in the forward computation; thus, our method is slower than the low-rank parameterization. However, the computation time is not a dominant factor in practical FL scenarios as shown in Table 7, but rather the communication cost takes a majority of the total training time. It is evidenced by the fact that FedPara offers better Pareto efficiency than all compared methods because our method has higher accuracy than low-rank approaches and much less training time than the original one.
>
>
> We have made the discussion detailed (See the last paragraph of Section 5 on page 9) in the revision.

---

> > ### Comment · Reviewer_3Rte · 2021-12-06
> > **Thank you for your response**
> >
> > Thank you for your response!

---

> > > ### Author Response · Authors · 2021-12-06
> > > **Response to Reviewer 3Rte's response**
> > >
> > > We appreciate the reviewer's constructive comments and time again. If there is any remaining question, we will try our best to answer.

---

### Official Review · Reviewer_yBc3 · 2021-11-02

**Correctness:** 3
**Technical Novelty And Significance:** 2
**Empirical Novelty And Significance:** 3
**Recommendation:** 6
**Confidence:** 4

**Main Review:**

======================= After Rebuttal ======================

I have read the authors' responses. Part of my concerns is alleviated. I would like to increase my overall evaluation to 6.

========================================================

Strengths:
This paper is well organized and easy to follow. The authors provided comprehensive experiments.

Questions:
1. As for Figure 1, does the low-rank parameterization $W=XY^{\rm T}$ always leads to the Hadamard product of two low-rank inner matrices $W=W_1\circ W_2=(X_1Y_1^{\rm T})\circ(X_2Y_2^{\rm T})$? What if the low-rank weight matrix $W$ cannot be decomposed into the Hadamard product of two low-rank inner matrices?
2. Proposition 1 only provides the upper bound for the rank(W). It does not necessarily mean that $rank(W)=r_1r_2$. How could the constructed matrix always span a full-rank matrix when $r_1r_2\ge \min(m,n)$?
3. In the numerical results with varying $\gamma\in[0.1,0.9]$, what is the corresponding number of parameters?
4. The accuracy differences between the low-rank method and FedPara in the non-IID case are smaller than the differences in the IID case. Is that any explanation for this phenomenon?
5. It would be better to move the details of proof to the appendix and add more discussions in the main manuscript.

**Summary Of The Paper:**

This paper proposes a communication-efficient parameterization for federated learning by using low-rank weights. Several numerical experiments are conducted to demonstrate the effectiveness of the proposed FedPara and pFedPara.

**Summary Of The Review:**

The paper is mostly sound, while the contributions are marginally significant.

---

> ### Author Response · Authors · 2021-11-13
> **Response to Reviewer yBc3's Review**
>
> We thank the reviewer for the comments and the suggestion. We address the concerns and the questions below:
>
> **Q1. As for Figure 1, does the low-rank parameterization $W=XY^\top$ always leads to the Hadamard product of two low-rank inner matrices $W=W_1\odot W_2=(X_1Y_1^\top)\odot(X_2Y_2^\top)$ ? What if the low-rank weight matrix  cannot be decomposed into the Hadamard product of two low-rank inner matrices?**
>
> The low-rank parameterization and our parameterization are different weight modeling approaches that are not necessarily related to each other.  In our paper, we do not decompose the weights, but the parameterized network is trained in the factorized form as it is.
>
> While it does not affect our conclusion, it is an interesting question whether any Hadamard product decomposition of a low-rank weight exists. We could not find any result for the question in our further survey, and we have not investigated yet (it would be worthwhile to investigate in the future).
>
>
> **Q2. Proposition 1 only provides the upper bound for the rank(W). It does not necessarily mean that $rank(W)=r_1r_2$. How could the constructed matrix always span a full-rank matrix when $r_1r_2>=min(m,n)$?**
>
> We guess the reviewer left this comment after reading this sentence, “the constructed matrix can span a full-rank matrix” in our paper. We revise and clarify the sentence to avoid similar misunderstandings in Section 2.2 (last paragraph of page 3) as “the constructed matrix does not have a low-rank restriction and is able to span a full-rank matrix with a high chance (see Fig. 6)”.
>
> From our initial submission, we have NEVER mentioned “always,” and our intention was NOT “always”. Our parameterization does not have low-rank constraints and is able to construct a full-rank matrix with high frequency (refer to Fig. 6 for the empirical evidence). In contrast, the traditional low-rank parameterization can NEVER span a full-rank matrix.
>
>
> **Q3. In the numerical results with varying $\gamma \in [0.1, 0.9]$, what is the corresponding number of parameters?**
>
> In Figure 4 on page 7, we adjusted the number of parameters by changing $\gamma$. The below tables are $\gamma$'s with their corresponding numbers of parameters.
>
>
>           VGG16 (CIFAR-10)
>         γ       No. parameters
>     =================================
>       0.1          1.55M
>       0.2          2.33M
>       0.3          3.31M
>       0.4          4.45M
>       0.5          5.79M
>       0.6          7.33M
>       0.7          9.01M
>       0.8          10.90M
>       0.9          12.92M
>     =================================
>      original      15.25MN
>
>           VGG16 (CIFAR-100)
>         γ       No. parameters
>     =================================
>       0.1          1.59M
>       0.2          2.38M
>       0.3          3.36M
>       0.4          4.50M
>       0.5          5.84M
>       0.6          7.38M
>       0.7          9.05M
>       0.8          10.94M
>       0.9          12.96M
>     =================================
>      original      15.30MN
>
> We have added the above table in Section C.2 of the supplementary material (See Table 5 on page 18).
>
> **Q4. The accuracy differences between the low-rank method and FedPara in the non-IID case are smaller than the differences in the IID case. Is that any explanation for this phenomenon?**
>
> As shown in Table 2 on page 5, the tendency the reviewer mentioned is more significant in the RNN experiment. For Shakespeare, the data distribution of the non-IID case is more heterogeneous than the non-IID settings of image recognition datasets.  This tendency is observed according to the degree of the data heterogeneity.
>
> The prior work [B1] shows that the convergence slows down in non-IID data settings, compared to IID settings. In this regard, many prior works have been proposed to increase the convergence stability in non-IID data distribution.
>
> From this evidence, we think that the difficulty of heterogeneous data leads to a smaller gap in the non-IID settings than the IID cases due to the difficulty of learning on non-IID data settings.
>
> [B1] Li, Xiang and Huang, Kaixuan and Yang, Wenhao and Wang, Shusen and Zhang, Zhihua, On the convergence of fedavg on non-iid data, ICLR, 2020, https://arxiv.org/abs/1907.02189
>
>
> **Q5. It would be better to move the details of proof to the appendix and add more discussions in the main manuscript.**
>
> We appreciate the comment to improve the paper. We have reflected the reviewer's comment in the revision (See Section A.1 on pages 14-15).

---

### Official Review · Reviewer_5RoZ · 2021-11-02

**Correctness:** 3
**Technical Novelty And Significance:** 2
**Empirical Novelty And Significance:** 2
**Recommendation:** 6
**Confidence:** 4

**Main Review:**

======================= Post Rebuttal ======================

After reading the authors' response. Part of my concerns are addressed. Thus
I would like to increase my overall evaluation to 6. And I believe that this work
can make a decent contribution to the efficient FL training area.

========================================================


The paper proposes to train low rank factorized model on the client side to improve the communication efficiency of federated learning while introducing a novel low rank factorization method.

Pros:
1. The paper is well written, and the research direction of enhancing the communication efficiency of federated learning is potentially impactful.
2. The proposed low rank factorization method is novel.
3. Both theoretical analysis and empirical results are provided to justify the effectiveness of FedPara.

Cons:
1. The main concern I have for the current version of the draft is that the proposed FedPara method is only compared against the traditional low rank based model factorization method, e.g., vanilla low rank model or Pufferfish. It is not clear if FedPara is also better compared against other communication efficient federated learning methods, e.g., [1, 2] or recently proposed low rank based communication efficient FL method [3].
2. By looking at Algorithms 1 and 2, it is not clear which weights are kept on the client side and which weights are kept on the server side. It says “Optimize(W)”, does it mean only $X_1, Y_1$ and $X_2, Y_2$ are optimized? Also for aggregate $X_1, Y_1$, and $X_2, Y_2$, does it require to transform them back to W before aggregation across clients’ updates? The authors are supposed to clarify this.
3. How to choose $\gamma$ is not clear, a fixed rank ratio seems to make sense. But it may also be possible that different layers can contain different redundancy. Is there a chance to adaptively choose $\gamma$ for different layers?
4. It is recently observed that the model weights of the Transformer are not low rank. Does it mean FedPara can not be applied over the Transformer architectures?

[1] https://arxiv.org/pdf/1610.05492.pdf

[2] http://proceedings.mlr.press/v108/reisizadeh20a/reisizadeh20a.pdf

[3] https://arxiv.org/pdf/2104.12416.pdf

[4] https://openreview.net/forum?id=_sSHg203jSu

**Summary Of The Paper:**

This paper introduces FedPara, a low-rank based method for achieving communication efficient federated learning. On the client-side, low rank factorized models are trained. Different from the traditional approaches to finding low rank factorized networks, FedPara introduces a novel low rank factorization strategy, which attains a higher maximal rank for the weight matrices. The paper also identifies that FedPara can be used to enhance personalized federated learning (by introducing a variant of FedPara, i.e., pFedPara). Both theoretical analysis and experimental results are provided to show the effectiveness of FedPara.

**Summary Of The Review:**

Overall I think the paper introduces a valid research idea on improving the communication efficiency of federated learning. However, several concerns exist (please refer to “Cons” in “Main Review”). If my major concerns are addressed, I will consider increasing my overall evaluation score.

---

> ### Author Response · Authors · 2021-11-13
> **Response to Reviewer 5RoZ's Review**
>
> We thank the reviewer for the time and their thorough review. We address the concerns and the questions below:
>
> **Q1. It is not clear if FedPara is also better compared against other communication efficient federated learning methods.**
>
> We discussed [A1, 2] in the initial submission, and we have added [A3] in the related work section (on page 9) of this revision. Thank the reviewer for introducing the paper based on the low-rank approach.
>
> Our low-rank baseline is actually motivated by the low-rank structured update at [A1]. In this rebuttal, we additionally evaluate with FedPAQ [A2]. FedPAQ [A2] applies the quantization method only when uploading the model from clients to the server. Since we train the pre-factorized model from scratch, our parameterization can be combined with any quantization method, including FedPAQ.
>
> To compare FedPAQ and FedPara, we train VGG16 on the CIFAR-10 IID setting and quantize 32 bits floating-point to 16 bits floating points. The below table shows the accuracy and transferred size per round:
>
>
>     Model              Acc.     Transferred size per round
>     =====================================================
>     FedAvg             83.68    122 MB
>     FedPAQ             82.67    91.4 MB
>     -----------------------------------------------------
>     FedPara (γ=0.5)    83.82    46.4 MB
>     + FedPAQ           83.58    34.74 MB
>     =====================================================
>
> FedPara ($\gamma=0.5$) achieves higher accuracy than FedPAQ. FedPAQ might lose the updated weight information due to the quantization, whereas FedPara can transfer the full information of updated weights to the server. Moreover, FedPara reduces transferred bits per round by 50% compared to FedPAQ. FedPara reduces both uplink and downlink communication costs, but FedPAQ only reduces uplink costs.
>
> FedPAQ achieves 1.01% lower accuracy and 25% lower communication costs than FedAvg. FedPara with quantization achieves 0.24% lower accuracy and transfers 25% lower communication costs than FedPara. This consistency shows that our method is compatible with the quantization algorithm.
>
> We have added these additional experiments in Section D.3 of the supplementary material (See pages 22-23).
>
> [A1] Konecny, Jakub and McMahan, H Brendan and Yu, Felix X and Richtarik, Peter and Suresh, Ananda Theertha and Bacon, Dave, Federated learning: Strategies for improving communication efficiency, arXiv, 2016
>
> [A2] Reisizadeh, Amirhossein and Mokhtari, Aryan and Hassani, Hamed and Jadbabaie, Ali and Pedarsani, Ramtin, Fedpaq: A communication-efficient federated learning method with periodic averaging and quantization, AISTATS, 2020
>
> [A3] Qiao, Zhefeng and Yu, Xianghao and Zhang, Jun and Letaief, Khaled B, Communication-Efficient Federated Learning with Dual-Side Low-Rank Compression, arXiv, 2021
>
> **Q2. By looking at Algorithms 1 and 2, it is not clear which weights are kept on the client side and which weights are kept on the server side. It says “Optimize(W)”, does it mean only $X_1, Y_2$ and $X_2, Y_2$ are optimized? Also for aggregate $X_1, Y_2$ and $X_2, Y_2$ does it require to transform them back to W before aggregation across clients’ updates?**
>
> Thanks for the comment. We have revised the algorithms to clarify the confusion in Section C.3 (See pages 18-19).
>
> Note that $X_1, Y_2$ and $X_2, Y_2$ are directly used in all the stages. The server aggregates $X_1, Y_2$ and $X_2, Y_2$ directly without re-composing W. While $X_1, Y_2$ and $X_2, Y_2$ are communicated between clients and sever in the FedPara scenario, only a side of low-rank matrices $X_1, Y_2$ in pFedPara is transferred to the server.
>
>
> **Q3. How to choose $\gamma$ is not clear, a fixed rank ratio seems to make sense. But it may also be possible that different layers can contain different redundancy. Is there a chance to adaptively choose for different layers?**
>
> As mentioned in Section 3.1, $\gamma$ can be tunned for each layer by layer differently in the model. $\gamma$ can be regarded as hyperparameters. While it is off the scope of this work, it would be interesting to see that recent advances of neural architecture search or hyper-parameter search methods may be applied to further improve the performance.
>
> We have revised to emphasize the related paragraph more specifically (See Sec. 3.1 and page 5).
>
> **Q4. It is recently observed that the model weights of the Transformer are not low rank. Does it mean FedPara can not be applied over the Transformer architectures?**
>
> Our key contribution is that our parameterization does not have low-rank restrictions according to Propositions 1 and 3. We believe that our method may be applied to the Transformer case, and ours would be likely to be better than the traditional low-rank counterpart. While our parameterization can be generalized to the Transformer architectures like CNN and RNN models shown in this submission, we leave the experiment for future research due to the limited period of rebuttal.

---

> > ### Comment · Reviewer_5RoZ · 2021-12-05
> > **Thanks for the rebuttal**
> >
> > After carefully reading the rebuttal, my major concerns are resolved. I would like to increase my overall evaluation score from 5 to 6.

---

> > > ### Author Response · Authors · 2021-12-06
> > > **Response to Reviewer 5RoZ's update**
> > >
> > > Thank the reviewer for the helpful comments and for raising the score. We are happy to hear that the major concerns are resolved. If the reviewer has any remaining concerns, we are glad to discuss them.

---

### Author Response · Authors · 2021-11-29
**Revision Summary**

We appreciate the valuable reviews that help to improve our paper. We summarize the revision as follows:

- Adding the comparison of other communication efficient federated learning method ($\color{red}{Reviewer\ 5RoZ}$, $\color{orange}{Reviewer\ M7KC}$) $\rightarrow$ Section D.3 of the supplementary material (pages 22-23)
- Fixing algorithms clearer ($\color{red}{Reviewer\ 5RoZ}$) $\rightarrow$ Section C.3 of the supplementary material (pages 18-19)
- Adding the discussion of the rank hyperparameter $\gamma$ ($\color{red}{Reviewer\ 5RoZ}$) $\rightarrow$ Section 3.1 (pages 5)
- Fixing and clarifying the sentence about no low-rank constraints of our method ($\color{blue}{Reviewer\ yBc3}$) $\rightarrow$ Section 2.2 (page 3)
- Clarifying the number of parameters corresponding to $\gamma$ ($\color{blue}{Reviewer\ yBc3}$) $\rightarrow$ Section C.2 of the supplementary material (Table 5 on page 18)
- Moving the details of proof to the appendix ($\color{blue}{Reviewer\ yBc3}$) $\rightarrow$ Section A.1 of the supplementary material (pages 14-15)
- Adding discussion of the key limitation in terms of expressiveness ($\color{green}{Reviewer\ 3Rte}$) $\rightarrow$ Section 5 (pages 9)
- Adding comparison between our and low-rank parameterization ($\color{green}{Reviewer\ 3Rte}$) $\rightarrow$ Section 5 (pages 9)
- Adding the final accuracy during longer rounds and the consistent tendency ($\color{orange}{Reviewer\ M7KC}$) $\rightarrow$ Section D.2 of the supplementary material (Table 9 on page 21)

We believe that our submission becomes more convincing with constructive feedback. Other concerns not included in the revision are addressed and discussed in the respective comments. We will try to address more feedbacks, if any, in the final version if accepted.

---

### Decision · Program_Chairs · 2022-01-20

**Decision:**

Accept (Poster)

**Comment:**

Dear Authors,

The response you have provided, based on the main concerns of reviewers, have answered most of the questions raised.
As far as I understand from the added experiments you have provided, the proposed methodology shows resilience in being at least as good as state of the art approaches, while at the same time it is a mathematically interesting approach.

Your response has covered concerns like comparison to other communication techniques (the comparison list is not complete, but yet your effort is appreciated), adding discussion on the rank parameter, add comments on expressiveness and the connection with low-rank parameterization, etc.

These efforts cannot be overlooked, and for that reason I suggest acceptance (poster).

Best

AC